# Rare earth elements distribution in the river sediments of Ditrău Alkaline massif, Eastern Carpathians

**Valentin Nicolae Coţac**[1], **Ovidiu Gabriel Iancu**[1], **Nicuşor Necula**[1,2]*, **Marius Cristian Sandu**[1], **Aurelia Andreea Loghin**[1], **Ovidiu Chişcan**[3], **George Stoian**[4]

**1** Department of Geology, Faculty of Geography and Geology, Alexandru Ioan Cuza University of Iasi, Iaşi, Romania, **2** Tulnici Research Center, Alexandru Ioan Cuza University of Iasi, Iaşi, Romania, **3** Faculty of Physics, Alexandru Ioan Cuza University of Iasi, Iaşi, Romania, **4** National Institute of Research and Development for Technical Physics, Iaşi, Romania

* nicusor.necula@uaic.ro

**Data Availability Statement:** All relevant data are within the manuscript.

**Funding:** NN and AAL are grateful to Romanian Ministry of Research, Innovation and Digitization,

## Abstract

Ditrău Alkaline Massif is one of the few syenitic Massifs in Europe subjected to mining exploration in the past, located in the Eastern Carpathians, Romania. The heterogenous petrography includes acid to ultrabasic rocks such as syenites, hornblendites, and diorites, making it the defining feature of the Massif. In this study, we analyze the river bed sediments of two rivers, Ditrău and Jolotca, draining the Ditrău Alkaline Massif to determine their geochemical composition, with particular interest in Rare Earth Elements. The analysis was carried out with various analytical methods, including Inductive Coupled Plasma Mass Spectrometry, powder X-ray diffractometry, and electronic microscopy for mineralogical analysis to determine the presence of heavy minerals and quantify the concentration of Rare Earth Elements in the river sediment samples. The results indicate the existence of heavy minerals and Rare Earth Elements in bearing minerals such as Monazite and Epidote. High concentration values of Light Rare Earth Elements are identified, with values more than double compared to the Upper Continental Crust in some cases, of which stands out Cerium with 175.47 mg·kg⁻¹ and Lanthanum with 108.32 mg·kg⁻¹. Most samples share three main minerals: Quartz, K Feldspar, and Albite, while Diopside is only present in the Jolotca sediment samples, and Plagioclase exists in Ditrău samples. Moreover, many identified trace elements, such as Niobium, Tantalum, and Zirconium, indicate high enrichments, with samples' mean value of 265.62 mg·kg⁻¹ for Zirconium and 200.24 mg·kg⁻¹ for Niobium. The sum of Rare Earth Elements identified in the analyzed river sediments is 385.01 mg·kg⁻¹ for Ditrău samples and 368.72 mg·kg⁻¹ for Jolotca, with Cerium being the most significant element. The La/Th and Hf distinction plots suggest a mixed felsic/basic source for the Ditrău area and an acidic source for the Jolotca area.

within Program 1 – Development of the national RD system, Subprogram 1.2 – Institutional Performance – RDI excellence funding projects, Contract no.11PFE/30.12.2021, for financial support. NN has used the computational facilities given by the infrastructure support from the Operational Program Competitiveness 2014–2020, Axis 1, under POC/448/1/1 Research infrastructure projects for public R&D institutions/Sections F 2018, through the Research Center with Integrated Techniques for Atmospheric Aerosol Investigation in Romania (RECENT AIR) project, under grant agreement MySMIS no. 12732.

**Competing interests:** The authors have declared that no competing interests exist.

## 1. Introduction

River bed sediment geochemistry improves the understanding of the geological context, the weathering processes, and host rock composition. It may offer significant information about the tectonic setting and the possible evolution of the continental crust [1]. They are also crucial for other activities, from mineral prospection and exploration to environmental assessment of natural and anthropogenic hazards [2]. Rare Earth Elements (REE) are a group of elements whose chemical properties gradually change with their decreasing ionic radii across the lanthanide series (lanthanide contraction), from Lanthanum (La) to Lutetium (Lu), causing slightly different behavior for Light REE (LREE) and Heavy REE (HREE) during dissolution, precipitation and adsorption [3–5]. They are industrially crucial due to their use in modern products such as cutting-edge technology, agricultural fertilizers, and medical techniques, and also widely used as tracers in geosciences [6,7]. Separation and partial partitioning of them leads to different rock-forming minerals that have significant implications for the geochemistry of the rocks [8]. They have also been used to trace the provenance of sediments [9], infer environmental change [10,11], and understand processes at the Earth's surface [12,13].

Generally, REE are characterized by strong partitioning into the particulate phase, coherent behavior during weathering, erosion, and fluvial transportation, and high resistance to chemical mobilization [14]. Light REE enrichment may be a good tracer when using the REE composition to identify the sources of terrestrial materials, compared to middle REE enrichment, which should cautiously be used considering diagenetic modification [15]. Close to the lithogenic source, the Ce anomaly tends to be much smaller, and the shale normalized $Nd_n/Yb_n$ ratio is much higher than that of the open ocean. However, the dissolved REE pattern may also reveal specific characteristics of the geological nature of the weathered lithogenic matter [16–20].

River sediments naturally sample and mediate large areas of eroded continental crust [21], and REE are generally less mobile and minimally fractionated during source-to-spill processes [9]. REE in river sediments thus represent their average value of alteration residues over a large area with different source rocks [4]. Therefore, river sediments usually exhibit a uniform REE pattern comparable to the Upper Continental Crust (UCC), characterized by a distinct enrichment of LREE (La to Eu) with weak or absent Ce and Eu anomalies [22]. In mining areas, several important factors control the geochemical behavior of REE, such as the primary lithogenic mineralogy of the solid waste, weathering or/and oxidation reactions, secondary phase mineralogy, sorption and desorption reactions, environmental management and climatic conditions [23–26]. As to [4], river sediments with REE contamination generally show REE patterns different from that of the pristine ones, e.g. [6,27], and concentrations of La > 80 mg·kg$^{-1}$, Ce > 100 mg·kg$^{-1}$, Nd > 85 mg·kg$^{-1}$, and Gd > 8.12 mg·kg$^{-1}$ in sediments are believed to be the REEs contributed from anthropogenic sources [6,28].

Sediments are the geological archives that record and preserve the signatures of the geological events that affect the source areas. Physical and chemical processes modify this fingerprint signal during transport and deposition [29]. All river bed sediments share the same evolution, from erosion, transportation, and deposition, preserving the chemical footprint of the host rock. The chemical footprint is a variable affected by the host rock's mineralogical composition, type of weathering, erosion, and sedimentation, making river bed sediments good tracers [2,30].

The geochemistry of river bed sediments research is widely used as a prospecting tool in different challenging regions, giving information about environmental impact and pollution [31–35]. Moreover, based on sediment mineralogy and geochemistry, the composition of the host rock can be determined and gives insights into the morphological and hydrological nature

of the basin and climate [36]. The chemical composition of river sediments also depends on many natural factors, including lithology, mineralogy, weathering potential of rocks in the catchment area, temperature, water pH, dissolved oxygen, content of organic matter, precipitation, river channel geometry, and bioturbation recycling, and sorting processes, as well as changes during diagenesis and metamorphism [1]. Human activity can profoundly impact sedimentation rates and the pollution of river sediments, especially in densely populated, urbanized regions. Increased erosion rates caused by intensified land use in the catchment due to mining and mineral processing, deforestation, or agriculture, for example, may restrict sediment transport because of excessive sediment loads. Another direct impact on sediment fluxes is caused by river engineering works such as dredging, river embankment, or dam and levee construction [37].

Unique by its genesis, chemistry, petrography, and mineralogy, the Ditrău Alkaline Massif (DAM) is one of the most exciting sites for geological investigations and especially in the geochemistry research for its REE and radioactive accumulations of the different minerals [38–52]. However, the geochemistry of river bed sediments has not yet been analyzed along the rivers that drain the Massif, an aspect that gives significant insights into the host rocks and the area's potential.

Our study aims to analyze the geochemistry of the river sediments focusing on the REE concentrations but also including the analysis of heavy minerals and trace elements. Naturally, the Massif is defined by significant REE concentrations, however, the influence of anthropic activities in the area was never really accounted for. Hence, the geochemical analysis of the riverbed sediments in the Ditrău and Jolotca rivers will provide significant information about the human impact on the environment. We expect higher concentrations of REE along the Jolotca River, which drains through areas subjected to mining activities in the past. We will also quantify the concentrations of REE in the river sediments and relate them to global standards such as the UCC to evaluate the possible enrichments of certain elements. Moreover, the analysis of the geochemical and mineralogical properties of the riverbed sediments will provide a better understanding of the geological source of the sediments within the DAM.

## 2. Study area

DAM is unique in Romania by size and petrography, having a metamorphic basement at the interior of the Eastern Carpathians, located in the proximity of the Gheorgheni, Lăzarea, and Ditrău (Fig 1). The DAM is an intermediate-size massif (about 800 km$^2$), exhibiting an excentric ring structure in which the more basic rocks tend to lie to the west, with an arcuate zone of syenitic rocks extending from the far north to the southeast and a large area dominated by nepheline syenite on the eastern side [53]. This alkaline intrusion structure includes REE [44,53] and other mineralizations such as niobium and molybdenum [54]. DAM origins in an extensional, rift-related continental intraplate setting at the southwestern margin of the East European Craton during the Upper Triassic [50,55–57]. In the following sequence, the area suffered several deformations from the middle of the Cretaceous to the Tertiary, associated with the Alpine tectonic events that concluded with the genesis of the nappe system [58,59].

Geologically, the igneous complex outcrops on the eastern side of the volcanic chain and defines the DAM. Petrographically, the Massif consists of various types of rocks, including the Neogene-Quaternary deposits, partially covered by andesitic pyroclastic deposits and lavas, and the Pliocene-Pleistocene sediments. The DAM interferes also with crystalline basement rocks of the Bucovinian Nappe inside the Eastern Carpathians [41,44,49].

The investigated area has a wide chemistry variation ranging from acid to ultrabasic rocks, which includes granitoids, diorites, and hornblendites [41]. The trait mentioned above is the

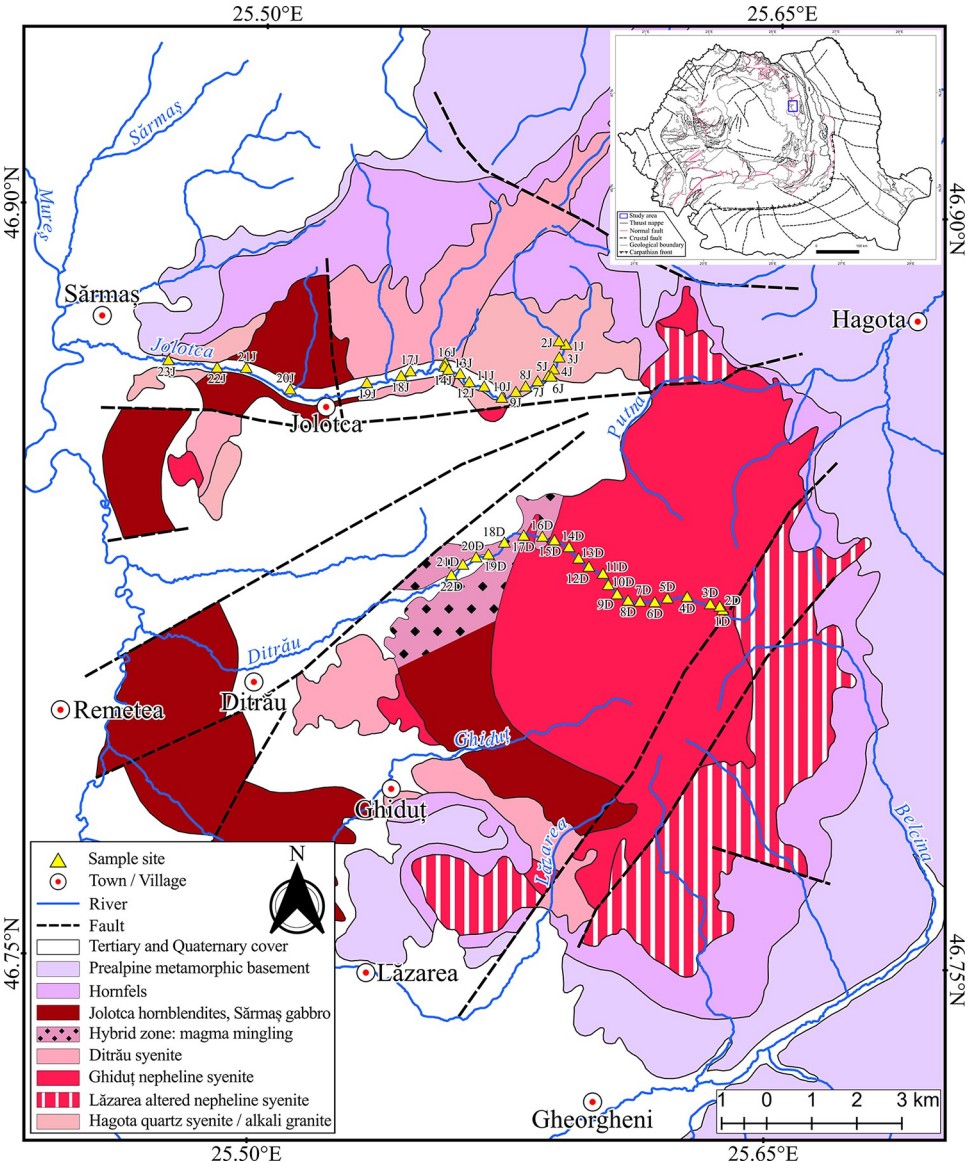

**Fig 1. Geological map of the DAM with the location of the collected samples.** In the inset, the geotectonic sketch of Romania. The map was created based on the information from various sources [44,47,59,60]. We used the color code for lithology from [61].

defining feature of the DAM, as the petrography of the area is highly heterogeneous and includes all the rock types: igneous, metamorphic, and sedimentary. Various authors proposed several models to describe the genesis of the region that consider it a multi-stage evolution phase. The most recent evolutionary model, to which many authors agree, was proposed by [44] as a four-stage process: (1) cca. 230 Ma.–the intrusion of the mafic and ultramafic rocks in the Jolotca region took place; (2) cca. 215 Ma.–the genesis of gabbro, diorites, monzodiorites, monzonites, syenites, and quartz syenites; (3) cca. 165–160 Ma.–the occurrence of nepheline syenites and formation of "Ditro essexites" followed by a series of dyke intrusions; (4) cca. 115 Ma.–final hydrothermal activity associated with the nappe transport due to tectonic uplift. These intrusions are supposedly related to the opening events of an ocean known as Meliata–

Hallstatt [55,62]. In these opening events, a model is proposed that rifting began somewhere in the middle Triassic in the Pelsonian substage [62].

Regarding the REE minerals identified in the Ditrău complex, they are part of the following six classes: REE(Y), Th, U–Carbonates; Nb, Ta, REE(Y), Ti, Zr, Th, Sn, U–Oxides; REE(Y)–Phosphates; REE(Y), Nb, Th, U, Zr, Pb, Ti–Silicates; Halides and Tellurides. They occur in small mineralization areas (Lăzarea, Hereb-Cianod, Ditrău Valley, Aurora, Putna, Creanga, and Halasag villages) and two prominent veins mineralization areas, Jolotca and Belcina [43,56,63].

The petrography of Ditrău Massif and their limits were thoroughly analyzed previously by [41,48,64,65] and finely synthesized by [44]. However, our study aims to investigate the geochemistry of the river bed sediments, especially the REE distribution, given that REE mineralizations are well-known in the area, but to this day, a detailed study on river sediments in the Massif has not yet been done.

DAM consists of three major geographical regions: the southern region of Lăzarea, the main area to the east of Ditrău village, and the northern region around the village of Jolotca [44]. The main area next to Ditrău village consists mainly of syenites and nepheline syenites, previously characterized as the "Ditro-essexites", which is a hybrid zone and igneous brecciation that consists of syenites and diorites [51]. In the northern area around the village of Jolotca, the characteristic petrography consists of mafic and ultramafic igneous rocks such as alkali gabbro and alkali diorite [44].

For this study, we analyze the geochemistry and mineralogy of river sediments along the Jolotca and Ditrău rivers (Fig 1). The rivers flow from east to west, passing through different rock types. Jolotca River drains the granitoid area further to the west through the syenite and the hornblendite deposits, respectively, as in the end to flow once again over the granitoid deposits while exiting the study area. The alkali granites are light gray with a light red hue; their principal mineral components are quartz, K-feldspar, plagioclase, biotite, and amphibole. The syenite area consists of syenite-monzosyenite [47], and their mineral composition includes feldspar, orthoclase, and microcline supplemented by rare minerals such as titanite, apatite, and zircon. The essential black hornblendite minerals include hornblende, titanite, biotite with plagioclase, apatite, and pyrite [65]. Ditrău River drains the northern sector of the area, passing through a large area that consists primarily of nepheline syenite, followed by the gabbro hybrid zone, and lastly, the Pliocene-Pleistocene sedimentary deposits. The syenite deposits are white and contain alkali feldspar, nepheline, sodalite, amphibole, and biotite, with secondary minerals such as zircon and monazite.

## 3. Materials and methods

To succeed with our study, we sampled the river bed sediments in several locations. Further on, we used different analytical methods to determine the samples' REE and mineral content. We generated distribution maps of the essential heavy, trace, and REE based on the samples' concentrations to better observe their fingerprint along the river channel and natural landscape.

### 3.1 Samples acquisition and preparation

We carried out the sampling surveys along the Jolotca and Ditrău rivers, totaling 45 sampling sites of river sediments: 23 for Jolotca and 22 for Ditrău, respectively. The sampling sites were chosen to cover all the petrographical units and the essential locations, such as the river confluences with their tributaries. The distance between points was not much considered due to the complex local geomorphology and the accessibility to the sampling point. In each location, we

cleared the area of debris and vegetation, after which we manually sampled the site with the help of shovels of different sizes and coring tools. We prepared two different sediment samples for every sampling point: a global one and another one panned and filtered to obtain a finer fraction for mineralogical analysis. Each sample weighed more than 1 kg and was stored in zip-lock bags.

In the laboratory, all the original samples were air-dried and sieved through six different sieves: 4 mm > 2 mm > 0.5 mm > 0.25 mm > 0.125 mm > 0.063 mm. We prepared two samples for every point, one for geochemical analysis and the other for mineralogical analysis. The fractions between 0.25 mm and 0.063 mm were used for the geochemical analysis. In contrast, the fractions between 1mm and 0.125 mm were used for the mineralogical samples, according to the literature and the accepted methodologies [66].

## 3.2 Analytical methods

For the geochemical analysis, we prepared ten samples for each river that were crushed to <75 μm in an agate mortar. The milled samples were analyzed in a certified lab (ALS Loughrea Co. Galway, Ireland), applying the ME-MS61L™ method (Super Trace Lowest DL 4A by ICP-MS) with add on method MS61L-REE™) based on four acid digestion: HNO3-HClO4-HF acid digestion, HCl leach (GEO-4A01) and ICP-MS. Quality Control/Quality Assurance included the analysis of sample duplicates (15% of total samples), blanks, and certified reference materials (EMOG-17, JK-17, MRGeo08, OREAS 210, OREAS 46, OREAS 905, OREAS-45e, OxA131. OxC129 and SY-4) with recovery ratios between 97% and 100% (ICP-MS).

A prepared sample (nominal weight of 0.25 g) is digested with 1.5 mL of concentrated nitric and perchloric acids, followed by concentrated hydrofluoric acid. The mixture is heated at 185°C until incipient dryness, leached with 50% hydrochloric acid, and diluted to volume with weak HCl. ICP-MS and ICP-AES then analyze the final solution, with results corrected for spectral inter-element interferences.

Ten samples were washed and air-dried for the mineralogical analysis. After that, the samples were separated with heavy fluid using diiodomethane with a density of 3.3 g/cm$^3$ to obtain a heavy mineral fraction and analyzed using SEM-EDX microscopy at the National Institute of Technical Physics in Romania. The samples with higher quantities of heavy minerals were crushed to <75 μm in an agate mortar and prepared for the powder XRD analysis. The X-ray powder diffraction analysis of the samples was performed at the Faculty of Physics, Alexandru Ioan Cuza University of Iasi, with a Shimadzu LabX XRD-6000 diffractometer. The semi-quantitative assessment was performed on the diffractograms using the XPowder 2010.01.10 software package, Match! version 2.4.7 [67], and QualX2.0 [68].

Another set of 10 samples was prepared for mineralogical analysis. Five global samples from all fractions and five samples from the 0.125 mm fraction were crushed to <75 μm in an agate mortar and sent for the powder XRD analysis. These samples were separated into two sub-samples, one global and one on the finer fraction, to determine the mineralogical species better.

## 3.3 Geostatistical analysis

To generate the distribution maps of the elements, we used the capabilities of the gstat R package [69,70]. R.stat [71] is an open-source software that allows users to easily manipulate spatial and temporal data, having more and more capabilities in the Geosciences community.

Hence, we used the ordinary kriging interpolation method to generate the distribution maps of the concerned elements along the river streams. To overcome the shortcomings and reduce the uncertainties given by the few samples available, for each element distribution map,

we ran 1000 conditional simulations based on the existing data and used their mean as the predicted value for the specific points along the river.

# 4. Results

## 4.1 Mineralogy analysis

The minerals identified with powder XRD and SEM-EDX analysis for Jolotca and Ditrău rivers (Table 1) consist mainly of quartz followed by K feldspar, albite, and diopside. Moreover, the powder XRD analysis also detected the presence of accessory minerals, such as titanite and rutile, in most of the samples and, in some cases, the presence of Titanium—bearing magnetite, hornblende, ferro-hornblende, zircon, augite, and actinolite (Table 1).

**Table 1. Main and accessory minerals identified with the powder XRD and SEM-EDX analysis.**

| | | Sample ID | | | | | | | | | |
| --- | --- | --- | --- | --- | --- | --- | --- | --- | --- | --- | --- |
| | | 2J | 6J | 10J | 14J | 21J | 1D | 6D | 16D | 19D | 22D |
| | | Jolotca River | | | | | Ditrău River | | | | |
| Main minerals | Qz | x | x | x | x | x | x | x | x | x | x |
| | K Fsp | x | x | x | x | x | x | x | x | x | x |
| | Pl | | | | | | x | x | x | x | |
| | Ab | x | x | x | x | x | x | x | x | x | x |
| | Cal | | | | x | | | x | x | | |
| | Di | x | x | x | x | x | | | x | | |
| | Act | | | | | | | | x | x | x |
| Accessory minerals | Fe-Hbl | | | | | | | | | x | x |
| | Ttn | x | x | x | x | x | x | x | x | x | x |
| | Zrn | x | x | x | | | x | | x | x | |
| | Ti-mag | x | x | x | | | | | | | |
| | Rt | | x | x | x | x | | | | | |
| | Amp | | x | x | | x | | | | x | |
| | Aug | | x | | x | x | | | | | |
| | Ap | | | | | | | | | | x |
| | Ep | | | | | | | | x | | |
| | Ccn | | | | | | | | x | | |
| | En | x | | | | | | | | | |
| | An | x | | | | x | | | | | |
| | Nph | x | x | | | x | | | | x | x |
| | Mnz | x | x | | x | | | | | | |
| | Dis-Ce | x | x | x | | x | | | | | |
| | Sdl | | x | | | | | | | x | x |
| | Aln-Ce | | x | x | | x | | | | | |
| | Pst-Ce | | | | x | | | | | | |
| | Xtm | | | | x | x | | | | | |
| | Ltn-Ce | | | | | x | | | | | |
| | Hap | | | | | x | | | | | |

The mineral symbols we used are in accordance with [72], as follows: Qz–Quartz, K Fsp–K feldspar, Pl—Plagioclase, Ab–Albite, Cal–Calcite, Di–Diopside, Act–Actinolite, Fe-Hbl–Ferro-Hornblende, Ttn–Titanite, Zrn–Zircon, Ti-mag–Titanium-bearing magnetite, Rt–Rutile, Amp–Amphibole, Aug–Augite, Ap—Apatite, Ep—Epidote, Ccn—Cancrinite, En—Enstatite, An—Anorthite, Nph—Nepheline, Mnz—Monazite, Dis-Ce—Dissakisite (Ce), Sdl—Sodalite, Aln-Ce—Allanite (Ce), Pst-Ce—Parisite (Ce), Xtm—Xenotime, Ltn-Ce—Lanthanite (Ce), Hap–Hydroxylapatite.

After the powder XRD analysis of the global samples 6J, 21J, 19D, and 22D, a fraction of the remaining material was separated with heavy fluid. The resulting fraction was sent to powder XRD analysis, identifying zircon, titanite, amphibole, augite, and actinolite.

After the separation with heavy fluid of the samples 1D, 2J, 6D, 10J, 14J, and 16D, grains were selected under the microscope and sent for SEM-EDX analysis, whose results show that most of the samples have zircon and titanite.

The analysis points out the existence of some accessory minerals specific to the DAM. These minerals were identified based on the powder XRD results with the help of Match! and QualX software, which identified minerals such as Cancrinite (Ccn), Lanthanite (Ce), Xeno-time (Xtm), Monazite (Mnz), Dissakisite (Ce), Parisite (Ce), Sodalite (Sdl), Allanite (Ce).

## 4.2 Geochemistry of major elements and trace elements

The chemical composition of major elements of the river sediments points out the content abundance of these elements in both rivers (Table 2), ordered as follows: Al (8.83%) > Fe (4.87%) > K (3.51%) > Na (3.03%) > Ca (2.46%) > Mg (1.55%) > Ti (1.13%) > Mn (0.14%) > P (0.08%) > S (0.02%). for Ditrău River, while for the Jolotca River, the content follows a

**Table 2. ICP-MS results of major elements content\* in the river sediments.**

| | Sample ID | Al | Ca | Fe | K | Mg | Na | P | S | Ti | ⴲMn |
|---|---|---|---|---|---|---|---|---|---|---|---|
| Jolotca River | 1J | 7.76 | 0.44 | 2.64 | 3.35 | 0.45 | 2.86 | 0.04 | 0.02 | 0.49 | 0.16 |
| | 4J | 7.86 | 0.49 | 2.89 | 3.28 | 0.5 | 3.12 | 0.05 | 0.01 | 0.55 | 0.16 |
| | 8J | 8.36 | 0.66 | 2.33 | 3.81 | 0.38 | 3.44 | 0.03 | 0.02 | 0.55 | 0.11 |
| | 11J | 7.88 | 1.36 | 3.04 | 2.87 | 0.62 | 3.15 | 0.06 | 0.02 | 0.86 | 0.13 |
| | 16J | 8.47 | 0.68 | 2.44 | 3.19 | 0.42 | 4.01 | 0.06 | 0.01 | 0.50 | 0.06 |
| | 18J | 7.93 | 1.08 | 3.12 | 2.93 | 0.51 | 3.53 | 0.06 | 0.02 | 0.81 | 0.13 |
| | 19J | 7.86 | 1.22 | 3.69 | 2.92 | 0.57 | 3.36 | 0.06 | 0.02 | 0.88 | 0.13 |
| | 20J | 6.99 | 5.55 | 9.65 | 1.47 | 2.29 | 2.84 | 0.26 | 0.07 | 2.20 | 0.17 |
| | 21J | 7.32 | 3.23 | 6.96 | 2.28 | 1.3 | 3.04 | 0.15 | 0.03 | 1.59 | 0.13 |
| | 23J | 7.53 | 3 | 5.37 | 2.33 | 1.28 | 3.19 | 0.15 | 0.03 | 1.31 | 0.12 |
| | Min | 6.99 | 0.44 | 2.33 | 1.47 | 0.38 | 2.84 | 0.03 | 0.01 | 0.49 | 0.06 |
| | Max | 8.47 | 5.55 | 9.65 | 3.81 | 2.29 | 4.01 | 0.26 | 0.07 | 2.20 | 0.17 |
| | Mean | 7.79 | 1.77 | 4.21 | 2.84 | 0.83 | 3.25 | 0.09 | 0.03 | 0.97 | 0.13 |
| Ditrău River | 1D | 10.3 | 0.32 | 3.87 | 6.19 | 0.35 | 2.71 | 0.02 | 0.01 | 0.47 | 0.09 |
| | 6D | 9.24 | 2 | 4.36 | 3.82 | 1.37 | 2.98 | 0.06 | 0.02 | 1.07 | 0.13 |
| | 9D | 8.57 | 2.11 | 4.6 | 3.41 | 1.67 | 3.01 | 0.07 | 0.02 | 1.02 | 0.13 |
| | 10D | 9.2 | 2.17 | 4.74 | 3.75 | 1.65 | 3.08 | 0.07 | 0.02 | 1.07 | 0.14 |
| | 15D | 8.99 | 2.09 | 4.43 | 3.61 | 1.59 | 3.11 | 0.07 | 0.02 | 0.95 | 0.14 |
| | 16D | 8.72 | 2.07 | 4.42 | 3.53 | 1.53 | 3.13 | 0.07 | 0.02 | 0.97 | 0.14 |
| | 17D | 8.88 | 2.3 | 4.57 | 3.28 | 1.6 | 3.14 | 0.08 | 0.02 | 1.02 | 0.15 |
| | 18D | 8.47 | 3.12 | 5.19 | 2.85 | 1.73 | 3.16 | 0.10 | 0.02 | 1.39 | 0.15 |
| | 20D | 8.18 | 3.8 | 5.69 | 2.51 | 1.9 | 3.13 | 0.13 | 0.02 | 1.47 | 0.16 |
| | 22D | 7.81 | 4.66 | 6.83 | 2.18 | 2.18 | 2.87 | 0.15 | 0.02 | 1.85 | 0.19 |
| | Min | 7.81 | 0.32 | 3.87 | 2.18 | 0.35 | 2.71 | 0.02 | 0.01 | 0.47 | 0.09 |
| | Max | 10.3 | 4.66 | 6.83 | 6.19 | 2.18 | 3.16 | 0.15 | 0.02 | 1.85 | 0.19 |
| | Mean | 8.83 | 2.46 | 4.87 | 3.51 | 1.55 | 3.03 | 0.08 | 0.02 | 1.13 | 0.14 |

\*—Content of the major oxides is showcased as % of the sample. The ICP-MS sample's nominal weight of 100 grams was halved for analysis purposes.

ⴲ—Mn was expressed in ppm originally, however, due to its high values will be reported in % and grouped with the rest of the major elements.

similar pattern: Al (7.79%) > Fe (4.21%) > Na (3.25%) > K (2.84%) > Ca (1.77%) > Ti (0.97%) > Mg (0.83%) > Mn (0.13%) > P (0.09%) > S (0.03%).

As seen in the case of the identified trace elements (Table 3), we analyzed many of these components, expressed in mg·kg⁻¹, which we grouped into six sub-groups for better management and interpretation.

**4.2.1 Alkali/Alkaline earth metals (Cs, Li, Rb / Ba, Be, Sr).** In the Ditrău River sediments, Cs vary from 2.4 mg·kg⁻¹ to 5.9 mg·kg⁻¹ with a mean of 4.7 mg·kg⁻¹, compared to the Jolotca sediment samples that range from 0.7 mg·kg⁻¹ to 2.6 mg·kg⁻¹ and have a mean of 1.7 mg·kg⁻¹. We noticed that in the samples from the Ditrău River, Cs has a higher concentration. Similarly, Li concentrations are higher on Ditrău River (Fig 2A), varying from 35.8 mg·kg⁻¹ to 125.5 mg·kg⁻¹ with a mean of 69.4 mg·kg⁻¹ for the Ditrău, and in Jolotca River, the concentration varies from 9.3 mg·kg⁻¹ to 31.4 mg·kg⁻¹ with a mean of 17.18 mg·kg⁻¹. For Rb, the values in Ditrău River samples range from 81.6 mg·kg⁻¹ to 185.5 mg·kg⁻¹ with a mean of 126.3 mg·kg⁻¹, and in the Jolotca River, they vary from 41.5 mg·kg⁻¹ to 172.5 mg·kg⁻¹ with a mean 110.5 mg·kg⁻¹.

The Ba content within the river sediments ranges from 590 mg·kg⁻¹ to 810 mg·kg⁻¹ with a mean of 711.7 mg·kg⁻¹ for Ditrău, while for the Jolotca, Ba has concentrations from 448 mg·kg⁻¹ to 850 mg·kg⁻¹ with a mean of 572.3 mg·kg⁻¹. Be has low concentrations, around 4 mg·kg⁻¹ for both rivers. Sr in the samples from Ditrău River varies from 340 mg·kg⁻¹ to 1005 mg·kg⁻¹ with a mean of 672.3 mg·kg⁻¹, and for the samples from Jolotca River vary from 153 mg·kg⁻¹ to 1235 mg·kg⁻¹ with a mean of 389.1 mg·kg⁻¹.

**4.2.2 Transitional metals (Ag, Cd, Co, Cr, Cu, Hf, Mo, Ni, Re, Sc, V, W, Y, Zn).** For both rivers, the sediments have a poor concentration of Ag, Cd, and Re with a mean under 1 mg·kg⁻¹. For Ditrău samples, Co varies from 5.1 mg·kg⁻¹ to 23.9 mg·kg⁻¹ with a mean of 16.2 mg·kg⁻¹, and for Jolotca samples, from 6.2 mg·kg⁻¹ to 29.1 mg·kg⁻¹ with a mean of 10.2 mg·kg⁻¹. Cr in the Ditrău River samples has values that vary from 8.9 mg·kg⁻¹ to 71.1 mg·kg⁻¹ with a mean of 50.6 mg·kg⁻¹, and for Jolotca sediment samples from 16.4 mg·kg⁻¹ to 46.5 mg·kg⁻¹ with a mean of 25.7 mg·kg⁻¹. Cu has values that vary from 4 mg·kg⁻¹ to 18 mg·kg⁻¹, with a mean of 9.2 mg·kg⁻¹ in the Ditrău River samples, in Jolotca River of 5.4 mg·kg⁻¹ to 13.9 mg·kg⁻¹ with a mean of 7.8 mg·kg⁻¹. Hf in the Ditrău and Jolotca rivers samples varies from 4 mg·kg⁻¹ to 11.5 mg·kg⁻¹ with a mean of 5.6 mg·kg⁻¹. Mo in both rivers has a low concentration with a mean under 1.5 mg·kg⁻¹. Ni has higher values in the Ditrău samples (Fig 2B) that vary from 7.7 mg·kg⁻¹ to 44.5 mg·kg⁻¹ with a mean of 34.5 mg·kg⁻¹ compared to Jolotca with values that vary from 11.5 mg·kg⁻¹ to 38.2 mg·kg⁻¹ with a mean of 16.6 mg·kg⁻¹.

Sc in both rivers has values ranging from 0.9 mg·kg⁻¹ to 13.8 mg·kg⁻¹, with a mean of 6.9 mg·kg⁻¹ for Ditrău and 6 mg·kg⁻¹ for Jolotca River. V in the samples from Ditrău River has values that vary from 54.1 mg·kg⁻¹ to 170.5 mg·kg⁻¹ with a mean of 109.8 mg·kg⁻¹, and in the Jolotca River, from 40.6 mg·kg⁻¹ to 206 mg·kg⁻¹ with a mean of 70.5 mg·kg⁻¹. W in the samples from Ditrău River has low values with a mean under 1.5 mg·kg⁻¹, while for Jolotca River, it has a mean of 3.5 mg·kg⁻¹. Y varies from 10.6 mg·kg⁻¹ to 44.9 mg·kg⁻¹ and a mean of 22.2 mg·kg⁻¹ in the samples from Ditrău River and from 21.4 mg·kg⁻¹ to 62.7 mg·kg⁻¹ with a mean of 31.4 mg·kg⁻¹ for Jolotca River. Zn has slightly higher values in the Ditrău River samples, ranging from 94.3 mg·kg⁻¹ to 143 mg·kg⁻¹ with a mean of 128.9 mg·kg⁻¹. Jolotca River's concentrations vary from 59.4 mg·kg⁻¹ to 141 mg·kg⁻¹, with a mean of 76.1 mg·kg⁻¹.

**4.2.3 Other metals/nonmetals (Bi, Ga, In, Pb, Sn, Tl / Se) and metalloids (As, Ge, Sb, Te).** Both rivers Bi, In, and Tl have low concentrations with a mean under 0.1 mg·kg⁻¹. For Ditrău River, Ga has values that vary from 25.4 mg·kg⁻¹ to 28.5 mg·kg⁻¹ with a mean of 27.2 mg·kg⁻¹, and for Jolotca, the concentration ranges from 21.3 mg·kg⁻¹ to 25.3 mg·kg⁻¹ with a mean of 22.8 mg·kg⁻¹. Pb shows values that vary from 8.2 mg·kg⁻¹ to 11.5 mg·kg⁻¹ with a mean

**Table 3. ICP-MS results of trace elements in mg·kg⁻¹ .**

| Sample ID | Alkali/Alkaline earth metals | | | | | | Transitional metals | | | | | | | | | | | | | | Other metals/nonmetals | | | | | | | Metalloids | | | | High-field strength element tracers | | | U and Th | |
|---|---|---|---|---|---|---|---|---|---|---|---|---|---|---|---|---|---|---|---|---|---|---|---|---|---|---|---|---|---|---|---|---|---|---|---|---|
| | Cs | Li | Rb | Ba | Be | Sr | Ag | Cd | Co | Cr | Cu | Hf | Mo | Ni | Re | Sc | V | W | Y | Zn | Bi | Ga | In | Pb | Sn | Tl | Se | As | Ge | Sb | Te | Zr | Nb | Ta | U | Th |
| **Jolotca River** | | | | | | | | | | | | | | | | | | | | | | | | | | | | | | | | | | | | |
| 1J | 2.67 | 19.3 | 165 | 450 | 4.79 | 153 | 0.02 | 0.06 | 7.67 | 22.4 | 8.76 | 4.16 | 0.99 | 14.7 | 0 | 5.35 | 46.3 | 8.5 | 21.7 | 62 | 0.42 | 22.5 | 0.04 | 11.8 | 4.36 | 0.48 | 0.1 | 9.6 | 0.17 | 0.37 | 0.02 | 186 | 124.5 | 9.86 | 5.69 | 40.7 |
| 4J | 2.65 | 22.6 | 172.5 | 448 | 5.03 | 170 | 0.01 | 0.06 | 8.31 | 24.2 | 8.78 | 4.78 | 1.02 | 14.9 | 0 | 5.2 | 48.8 | 7.76 | 25.3 | 70.6 | 0.4 | 25.1 | 0.03 | 12.5 | 4.83 | 0.46 | 0.13 | 10.05 | 0.15 | 0.31 | 0.03 | 220 | 143.5 | 10.4 | 6.27 | 38.5 |
| 8J | 2.22 | 31.4 | 157.5 | 520 | 4.21 | 305 | 0.01 | 0.05 | 6.26 | 17.6 | 5.4 | 4.67 | 0.86 | 11.55 | 0 | 3.44 | 40.6 | 3.89 | 21.4 | 59.4 | 0.15 | 24.4 | 0.04 | 11.2 | 3.78 | 0.33 | 0.07 | 4.07 | 0.16 | 0.27 | 0.01 | 241 | 163 | 9.89 | 7.12 | 34.8 |
| 11J | 2.13 | 26.7 | 116.5 | 570 | 3.69 | 434 | 0.02 | 0.09 | 9.29 | 29.9 | 6.88 | 5.61 | 1.4 | 16.8 | 0 | 5.78 | 66.6 | 3.46 | 29.7 | 75.4 | 0.16 | 23.1 | 0.05 | 12.9 | 3.75 | 0.29 | 0.15 | 4.16 | 0.21 | 0.31 | 0.01 | 268 | 189 | 10.95 | 9.3 | 38.1 |
| 16J | 1.96 | 13.1 | 138 | 490 | 5.53 | 280 | <0.002 | 0.03 | 6.89 | 16.4 | 6.28 | 4.02 | 0.65 | 12.15 | 0 | 3.98 | 42 | 4.23 | 24.7 | 59.6 | 0.1 | 25.3 | 0.04 | 12.45 | 3.99 | 0.37 | 0.06 | 1.53 | 0.17 | 0.21 | 0.01 | 169 | 137.5 | 9.94 | 8.19 | 35 |
| 18J | 1.78 | 16.5 | 120 | 540 | 4.34 | 345 | <0.002 | 0.06 | 7.95 | 20.8 | 5.61 | 4.56 | 1.16 | 11.95 | 0 | 4.68 | 54.2 | 3.87 | 28.8 | 65.3 | 0.12 | 22 | 0.05 | 12.5 | 4.3 | 0.31 | 0.09 | 2.69 | 0.19 | 0.25 | 0.01 | 226 | 203 | 13.65 | 7.56 | 40.8 |
| 19J | 1.69 | 16.5 | 108 | 590 | 3.76 | 373 | 0.01 | 0.06 | 8.31 | 24.5 | 7.69 | 4.86 | 1.23 | 13.4 | 0 | 5.1 | 61.6 | 3.07 | 29.5 | 67.9 | 0.21 | 21.3 | 0.05 | 14.6 | 4.11 | 0.29 | 0.1 | 2.55 | 0.2 | 0.3 | 0.01 | 240 | 192 | 12.6 | 8.3 | 39.7 |
| 20J | 0.72 | 9.3 | 41.5 | 850 | 2.05 | 1235 | 0.03 | 0.26 | 29.1 | 46.5 | 13.9 | 11.5 | 7.71 | 38.2 | 0 | 13.85 | 206 | 1.32 | 62.7 | 141 | 0.12 | 22 | 0.12 | 14.55 | 4.84 | 0.08 | 0.15 | 2.23 | 0.47 | 0.2 | 0.03 | 430 | 292 | 18.95 | 4.41 | 36.7 |
| 21J | 1.26 | 13.1 | 83.1 | 690 | 2.9 | 736 | 0.01 | 0.1 | 16.8 | 34.1 | 9.53 | 8.54 | 2.54 | 22.8 | 0 | 9.46 | 145 | 2.45 | 50 | 94.5 | 0.1 | 21.8 | 0.07 | 12.75 | 4.68 | 0.2 | 0.1 | 2.51 | 0.33 | 0.25 | 0.02 | 342 | 258 | 17.85 | 6.54 | 36.9 |
| 23J | 1.36 | 14.1 | 84.5 | 690 | 3.05 | 707 | 0.02 | 0.12 | 16.25 | 33.4 | 9.07 | 7.03 | 2.39 | 22.1 | 0 | 9.14 | 119.5 | 2.08 | 40.8 | 95.2 | 0.11 | 21.4 | 0.07 | 11.85 | 4.17 | 0.21 | 0.09 | 2.64 | 0.29 | 0.28 | 0.01 | 279 | 200 | 13.25 | 4.9 | 25.6 |
| Min | 0.72 | 9.3 | 41.5 | 448 | 2.05 | 153 | 0.01 | 0.03 | 6.26 | 16.4 | 5.4 | 4.02 | 0.65 | 11.55 | 0 | 3.44 | 40.6 | 1.32 | 21.4 | 59.4 | 0.1 | 21.3 | 0.03 | 11.2 | 3.75 | 0.08 | 0.06 | 1.53 | 0.15 | 0.2 | 0.01 | 169 | 124.5 | 9.86 | 4.41 | 25.6 |
| Max | 2.67 | 31.4 | 172.5 | 850 | 5.53 | 1235 | 0.03 | 0.26 | 29.1 | 46.5 | 13.9 | 11.5 | 7.71 | 38.2 | 0 | 13.85 | 206 | 8.5 | 62.7 | 141 | 0.42 | 25.3 | 0.12 | 14.6 | 4.84 | 0.48 | 0.15 | 10.05 | 0.47 | 0.37 | 0.03 | 430 | 292 | 18.95 | 9.3 | 40.8 |
| Mean | 1.73 | 17.18 | 110.53 | 572.39 | 3.79 | 389.2 | 0.01 | 0.07 | 10.3 | 25.71 | 7.89 | 5.63 | 1.49 | 16.6 | 0 | 6.02 | 70.56 | 3.53 | 31.42 | 76.17 | 0.16 | 22.85 | 0.05 | 12.67 | 4.26 | 0.27 | 0.1 | 3.47 | 0.22 | 0.27 | 0.02 | 251.04 | 183.98 | 12.39 | 6.66 | 36.4 |
| **Ditrău River** | | | | | | | | | | | | | | | | | | | | | | | | | | | | | | | | | | | | |
| 1D | 5.61 | 125.5 | 185.5 | 590 | 2.58 | 340 | 0.03 | 0.11 | 5.13 | 8.9 | 4.08 | 4.78 | 0.85 | 7.77 | 0 | 0.92 | 54.1 | 1.62 | 10.65 | 94.3 | 0.11 | 26.6 | 0.03 | 11.4 | 1.14 | 0.4 | 0.1 | 2.77 | 0.14 | 0.33 | 0.02 | 329 | 166 | 5.05 | 9.59 | 29 |
| 6D | 5.7 | 88.7 | 155 | 740 | 3.02 | 646 | <0.002 | 0.15 | 15.7 | 47.8 | 8.17 | 5.82 | 1.36 | 32.9 | 0 | 7.66 | 107 | 1.36 | 23.3 | 122.5 | 0.06 | 26.8 | 0.06 | 11.25 | 1.99 | 0.3 | 0.12 | 2.07 | 0.24 | 0.32 | 0.01 | 308 | 231 | 8.73 | 8.15 | 28.3 |
| 9D | 5.86 | 76.2 | 127 | 680 | 3.01 | 619 | 0.01 | 0.15 | 18.15 | 62.3 | 10.2 | 5.12 | 1.27 | 44.5 | 0 | 8.03 | 108.5 | 1.19 | 18.9 | 134 | 0.05 | 26.9 | 0.04 | 10.9 | 2.15 | 0.28 | 0.1 | 1.68 | 0.21 | 0.24 | 0.01 | 240 | 193.5 | 6.92 | 6.88 | 20.1 |
| 10D | 5.9 | 79.8 | 154.5 | 720 | 3.08 | 640 | 0.02 | 0.17 | 17.2 | 58.5 | 10.05 | 5.6 | 1.25 | 40.8 | 0 | 7.69 | 107.5 | 1.24 | 21 | 133 | 0.05 | 27.1 | 0.05 | 10.15 | 1.96 | 0.26 | 0.12 | 1.84 | 0.23 | 0.29 | 0.01 | 290 | 207 | 6.91 | 7.16 | 24.1 |
| 15D | 5.91 | 77 | 146 | 700 | 3.02 | 633 | 0.01 | 0.18 | 17.15 | 59.6 | 9.76 | 4.87 | 1.25 | 41.3 | 0 | 8.27 | 103.5 | 1.23 | 19 | 132.5 | 0.05 | 28.1 | 0.05 | 10.85 | 2.02 | 0.28 | 0.15 | 1.61 | 0.21 | 0.24 | 0.01 | 229 | 191 | 6.77 | 7.36 | 20.5 |
| 16D | 5.55 | 76 | 124 | 690 | 2.92 | 630 | 0.01 | 0.18 | 17.15 | 58 | 18 | 5.14 | 1.3 | 41 | 0 | 7.68 | 105 | 1.2 | 18.95 | 133.5 | 0.05 | 27.3 | 0.04 | 11.25 | 2.62 | 0.27 | 0.15 | 1.85 | 0.2 | 0.27 | 0.01 | 236 | 204 | 7.25 | 8.56 | 20.9 |
| 17D | 5.38 | 68.4 | 138.5 | 700 | 2.85 | 677 | <0.002 | 0.19 | 17.35 | 61.6 | 9.34 | 5.11 | 1.49 | 41.2 | 0 | 8.19 | 108.5 | 1.13 | 21.4 | 132.5 | 0.05 | 27.9 | 0.06 | 11.5 | 2.11 | 0.25 | 0.14 | 1.37 | 0.24 | 0.23 | 0.01 | 244 | 208 | 7.17 | 8.9 | 25.4 |
| 18D | 4.2 | 57.9 | 102.5 | 740 | 2.81 | 809 | <0.002 | 0.17 | 19.15 | 67.4 | 8.65 | 5.88 | 1.61 | 40.4 | 0 | 8.33 | 129.5 | 1.09 | 27.4 | 135.5 | 0.05 | 27.9 | 0.05 | 9.92 | 2.3 | 0.2 | 0.22 | 2.04 | 0.19 | 0.22 | 0.01 | 281 | 243 | 8.78 | 8.01 | 22 |
| 20D | 3.11 | 46.4 | 87.7 | 810 | 2.69 | 989 | <0.002 | 0.17 | 21.3 | 70.6 | 10 | 6.42 | 1.52 | 42.1 | 0 | 10.35 | 145.5 | 0.91 | 31.5 | 136 | 0.04 | 28.5 | 0.06 | 9 | 2.44 | 0.16 | 0.13 | 2.11 | 0.24 | 0.19 | 0.01 | 295 | 239 | 9.16 | 7.46 | 24.4 |
| 22D | 2.45 | 35.8 | 81.6 | 770 | 2.55 | 1005 | 0.01 | 0.2 | 23.9 | 71.1 | 9.55 | 8.82 | 1.69 | 43.6 | 0 | 12.9 | 170.5 | 0.92 | 44.9 | 143 | 0.04 | 25.4 | 0.1 | 8.22 | 2.96 | 0.13 | 0.13 | 1.69 | 0.29 | 0.2 | 0.01 | 387 | 313 | 13.5 | 9.43 | 31.3 |
| Min | 2.45 | 35.8 | 81.6 | 590 | 2.55 | 340 | 0 | 0.11 | 5.13 | 8.9 | 4.08 | 4.78 | 0.85 | 7.77 | 0 | 0.92 | 54.1 | 0.91 | 10.65 | 94.3 | 0.04 | 25.4 | 0.03 | 8.22 | 1.14 | 0.13 | 0.1 | 1.37 | 0.14 | 0.19 | 0.01 | 229 | 166 | 5.05 | 6.88 | 20.1 |
| Max | 5.91 | 125.5 | 185.5 | 810 | 3.08 | 1005 | 0.03 | 0.2 | 23.9 | 71.1 | 18 | 8.82 | 1.69 | 44.5 | 0 | 12.9 | 170.5 | 1.62 | 44.9 | 143 | 0.11 | 28.5 | 0.1 | 11.5 | 2.96 | 0.4 | 0.15 | 2.77 | 0.29 | 0.33 | 0.02 | 387 | 313 | 13.5 | 9.59 | 31.3 |
| Mean | 4.78 | 69.44 | 126.36 | 711.73 | 2.85 | 672.4 | 0.01 | 0.16 | 16.21 | 50.68 | 9.26 | 5.66 | 1.34 | 34.52 | 0 | 6.92 | 109.85 | 1.17 | 22.23 | 128.95 | 0.05 | 27.24 | 0.05 | 10.39 | 2.11 | 0.24 | 0.13 | 1.87 | 0.22 | 0.25 | 0.01 | 280.2 | 216.5 | 7.77 | 8.1 | 24.33 |

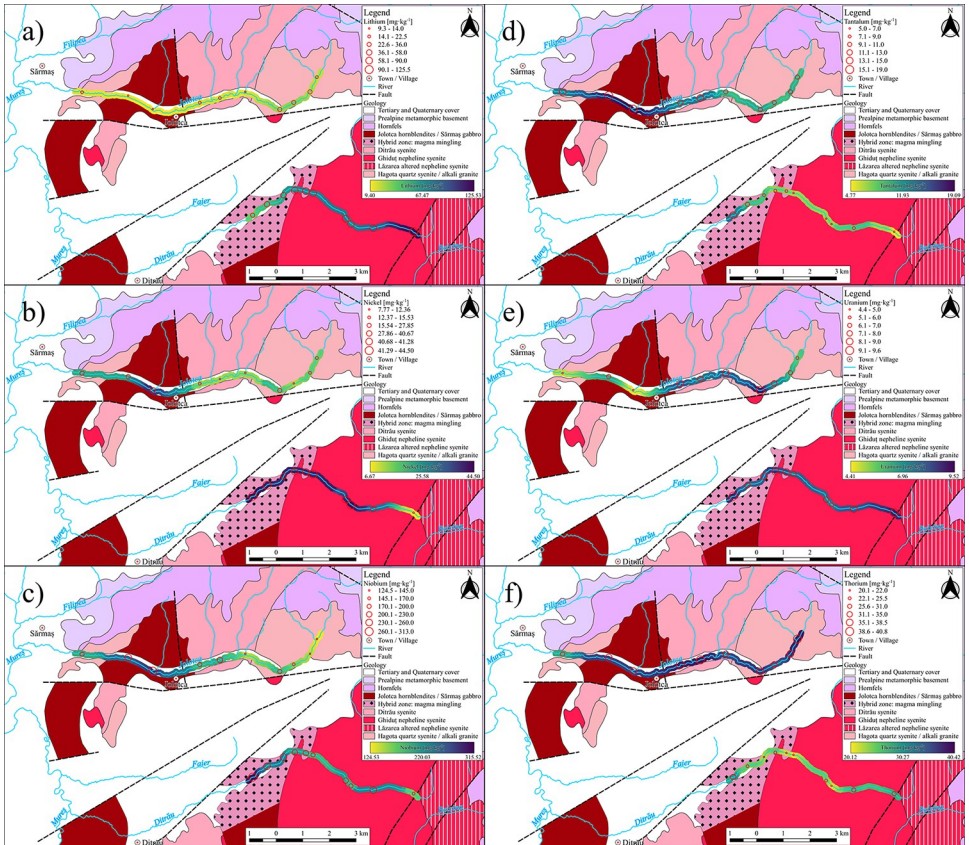

**Fig 2. Distribution maps of trace elements along the Ditrău and Jolotca river.** a) Lithium; b) Nickel; c) Niobium; d) Tantalum; e) Uranium; f) Thorium.

of 10.3 mg·kg$^{-1}$ for the samples from Ditrău, and for the samples from Jolotca River, from 11.2 mg·kg$^{-1}$ to 14.6 mg·kg$^{-1}$ with a mean of 12.6 mg·kg$^{-1}$. Sn has low concentrations for both rivers, with a mean under 2.5 mg·kg$^{-1}$ for Ditrău, and 4.2 mg·kg$^{-1}$ for Jolotca. Se has a mean value under 0.1 mg·kg$^{-1}$ in both river samples.

Regarding the Metalloids, As has low values, with a mean under 4 mg·kg$^{-1}$, and even lower values for Ge, Sb, and Te, with a mean under 1 mg·kg$^{-1}$ for both rivers.

**4.2.4 High field strength element tracers (Zr, Nb, Ta).** Both rivers show high concentrations of Zr, the minimum value being 169 mg·kg$^{-1}$ for the samples from Jolotca River and 229 mg·kg$^{-1}$ for the samples from Ditrău River. The maximum values are 430 mg·kg$^{-1}$ for the Jolotca River and 387 mg·kg$^{-1}$ for the Ditrău River. The mean values record 280.2 mg·kg$^{-1}$ for Ditrău and 251 mg·kg$^{-1}$ for Jolotca. Following the Zr trend, Nb shows, for both rivers, high concentrations (Fig 2C) with a minimum value for Ditrău River at 166 mg·kg$^{-1}$ and for Jolotca at 124.5 mg·kg$^{-1}$. The maximum value for Ditrău is 313 mg·kg$^{-1}$, and for Jolotca 292 mg·kg-1, the mean for Ditrău River is 216.5 mg·kg$^{-1}$, and for Jolotca 183.9 mg·kg$^{-1}$. Compared to Zr and Nb, Ta shows low concentrations (Fig 2D) with values ranging from 5 mg·kg$^{-1}$ to 13.5 mg·kg$^{-1}$ and a mean of 7.7 mg·kg$^{-1}$ for Ditrău River and Jolotca from 9.8 mg·kg$^{-1}$ to 18.9 mg·kg$^{-1}$ with a mean of 12.3 mg·kg$^{-1}$.

**4.2.5 U and Th.** In the sediment samples from Ditrău River, Th values vary from 20.1 mg·kg$^{-1}$ to 31.3 mg·kg$^{-1}$ with a mean of 24.3 mg·kg$^{-1}$, and in Jolotca River, from 25.6 mg·kg$^{-1}$ to 40.8 mg·kg$^{-1}$ with a mean of 36.3 mg·kg$^{-1}$. Compared to Th, U presents low concentrations

(Fig 2E and 2F) ranging from 6.8 mg·kg$^{-1}$ to 9.5 mg·kg$^{-1}$ with a mean of 8.1 mg·kg$^{-1}$ for Ditrău River and, for Jolotca, from 4.4 mg·kg$^{-1}$ to 9.3 mg·kg$^{-1}$ with a mean of 6.6 mg·kg$^{-1}$.

## 4.3 Rare earth elements

The chemical analysis of REE has significant variations for each element in terms of min, max, and mean within the samples (Table 4). For the Ditrău River, the sum of REEs (ΣREE) varies from 219 mg·kg$^{-1}$ to 632.8 mg·kg$^{-1}$ with a mean of 385 mg·kg$^{-1}$, and for Jolotca River, from 206 mg·kg$^{-1}$ to 1018.4 mg·kg$^{-1}$ with a mean of 368.7 mg·kg$^{-1}$. The samples have higher Light Rare Earth Elements (LREE) concentrations than Heavy Rare Earth Elements (HREE), as pointed out by the distribution maps as well (Fig 3). The ratio between LREE and HREE has a mean of 22 mg·kg$^{-1}$ for the Ditrău, while for Jolotca, the ratio has a mean of 14.9 mg·kg$^{-1}$. For Ditrău case, the sum of LREE varies from 211.2 mg·kg$^{-1}$ to 600 mg·kg$^{-1}$ with a mean of 368.1 mg·kg$^{-1}$, and for Jolotca River, from 190.6 mg·kg$^{-1}$ to 967.5 mg·kg$^{-1}$ with a mean of 345.3 mg·kg$^{-1}$. The sum of HREE for Ditrău samples varies from 7.8 mg·kg$^{-1}$ to 32.7 mg·kg$^{-1}$ with a mean of 16.7 mg·kg$^{-1}$, for Jolotca River from 15.3 mg·kg$^{-1}$ to 50.8 mg·kg$^{-1}$ with a mean of 23.1 mg·kg$^{-1}$.

Moreover, we calculated the Enrichment Factor (EF), which is often applied to evaluate the impacts of anthropogenic and lithogenic activities using several conservative elements (e.g., Al, Fe, and Sc) [73–76]. In our study, Fe was used as a conservative element to calculate the EF values of REE, as follows:

$$EF_i = (C_i/C_{Fe})_{sample}/(C_i/C_{Fe})_{UCC} \tag{Eq1}$$

where $EF_i$ represents the enrichment factor of the $REE_i$, and $C_i$ denotes the concentration of element "i" in the sediment samples (Table 5). The concentrations of Fe and REE in the UCC [77] were used as the background values, which were 31, 20.7, 32.3, 65.7, 6.3, 25.9, 4.7, 0.95, 2.8, 0.5, 2.9, 0.62, 2.3, 0.33, 1.5, 0.27 (unit: g/kg for Fe and mg·kg$^{-1}$ for REE) for Fe, Y, La, Ce, Pr, Nd, Sm, Eu, Gd, Tb, Dy, Ho, Er, Tm, Yb, and Lu, respectively. EF values were classified as per [78]: minimal (EF < 2), moderate (2 ≤ EF < 5), significant (5 ≤ EF < 20), very high (20 ≤ EF < 40), and extremely high (EF ≥ 40) enrichment. Moreover, an EF value below 1.5 indicated that REEs were mainly derived from natural processes, and an EF value over 1.5 indicated that REE were likely anthropogenic [79].

## 5. Discussions

Relating our results to the UCC [80] and other studies across Romania [81] and Europe [82], several noticeable aspects stand out for each category of elements.

## 5.1 Alkali/Alkaline earth metals

Compared to the UCC, all alkali/alkaline earth metals except Li and Sr have typical values. For Ditrău sediments, Li gets enriched compared to the Li/UCC ratio; in our area, Li enrichment is higher, with values from 1.7 to 5.9 with an average of 3.4. This aspect is particular for Ditrău River but not for Jolotca, which has values similar to UCC's. The Li enrichment in the Ditrău River sediments is related to the mineralogy of the syenites, which is known for not taking part in the feldspar structures, also confirmed by the presence of certain minerals, such as pyrochlore and biotite that have slightly high concentrations of Li [41]. Based on the distribution map (Fig 2), it can be seen that the high concentrations of Li overlap the syenites area.

Compared to the UCC, Sr shows values that vary from 1 to 3.1 with an average of 2.1, while for the samples from Jolotca River, Sr/UCC ratio presents values from 0.4 to 3.8 with an average of 1.4.

**Table 4. Rare earth element contents (mg·kg$^{-1}$) in Ditrău and Jolotca rivers.**

| Sample ID | La | Ce | Pr | Nd | Sm | Eu | Gd | Tb | Dy | Ho | Er | Tm | Yb | Lu | ΣREE | ΣLREE | ΣHREE | ΣLREE/ΣHREE | La$_n$/Yb$_n$ | La$_n$/Sm$_n$ | Gd$_n$/Yb$_n$ | Nd$_n$/Yb$_n$ | Eu/Eu* | Ce/Ce* |
|---|---|---|---|---|---|---|---|---|---|---|---|---|---|---|---|---|---|---|---|---|---|---|---|---|
| Ditrău River | | | | | | | | | | | | | | | | | | | | | | | | |
| 1D | 66.4 | 103.5 | 9 | 28 | 3.43 | 0.89 | 2.31 | 0.3 | 1.94 | 0.39 | 1.25 | 0.18 | 1.28 | 0.18 | 219.05 | 211.22 | 7.83 | 26.98 | 35.19 | 12.18 | 1.47 | 7.66 | 0.97 | 0.99 |
| 6D | 154.5 | 231 | 22.6 | 72.9 | 9.07 | 2.48 | 6.42 | 0.79 | 4.71 | 0.86 | 2.35 | 0.3 | 1.92 | 0.25 | 510.16 | 492.55 | 17.61 | 27.97 | 54.38 | 10.72 | 2.71 | 13.24 | 0.99 | 0.92 |
| 9D | 91.2 | 147.5 | 14.35 | 49.3 | 6.75 | 1.99 | 5.25 | 0.67 | 4.02 | 0.72 | 2.04 | 0.26 | 1.64 | 0.22 | 325.9 | 311.09 | 14.82 | 20.99 | 37.69 | 8.5 | 2.6 | 10.52 | 1.02 | 0.96 |
| 10D | 123 | 186 | 17.6 | 58.8 | 7.62 | 2.16 | 5.34 | 0.68 | 4.06 | 0.73 | 2.05 | 0.27 | 1.68 | 0.23 | 410.2 | 395.18 | 15.02 | 26.3 | 49.47 | 10.16 | 2.58 | 12.21 | 1.03 | 0.94 |
| 15D | 95.6 | 150.5 | 14.45 | 48.8 | 6.8 | 1.95 | 5.25 | 0.66 | 3.97 | 0.71 | 1.97 | 0.26 | 1.63 | 0.23 | 332.77 | 318.1 | 14.67 | 21.69 | 39.75 | 8.85 | 2.62 | 10.47 | 1 | 0.95 |
| 16D | 88.6 | 144.5 | 14.15 | 47.8 | 6.76 | 1.92 | 5.23 | 0.67 | 4.12 | 0.75 | 2.08 | 0.27 | 1.69 | 0.23 | 318.76 | 303.73 | 15.03 | 20.2 | 35.53 | 8.25 | 2.52 | 9.89 | 0.98 | 0.96 |
| 17D | 116 | 180 | 17.25 | 57.9 | 7.61 | 2.25 | 5.85 | 0.74 | 4.41 | 0.8 | 2.22 | 0.28 | 1.75 | 0.23 | 397.28 | 381.01 | 16.27 | 23.41 | 44.92 | 9.59 | 2.72 | 11.57 | 1.03 | 0.94 |
| 18D | 106 | 190 | 18.1 | 60 | 9.44 | 2.82 | 7.28 | 1 | 5.53 | 0.97 | 2.72 | 0.34 | 2.29 | 0.27 | 406.76 | 386.36 | 20.4 | 18.94 | 31.28 | 7.07 | 2.58 | 9.14 | 1.04 | 1.02 |
| 20D | 114.5 | 204 | 19.9 | 66.3 | 10.65 | 3.22 | 8.31 | 1.13 | 6.26 | 1.1 | 3.04 | 0.4 | 2.5 | 0.3 | 441.6 | 418.57 | 23.03 | 18.17 | 30.95 | 6.77 | 2.69 | 9.25 | 1.05 | 1 |
| 22D | 162 | 278 | 30.2 | 109.5 | 15.7 | 4.67 | 12.3 | 1.53 | 8.95 | 1.6 | 4.21 | 0.52 | 3.24 | 0.4 | 632.81 | 600.07 | 32.74 | 18.33 | 33.79 | 6.49 | 3.08 | 11.79 | 1.03 | 0.93 |
| Min | 66.4 | 103.5 | 9 | 28 | 3.43 | 0.89 | 2.31 | 0.3 | 1.94 | 0.39 | 1.25 | 0.18 | 1.28 | 0.18 | 219.05 | 211.22 | 7.83 | 18.17 | 30.95 | 6.49 | 1.47 | 7.66 | 0.97 | 0.92 |
| Max | 162 | 278 | 30.2 | 109.5 | 15.7 | 4.67 | 12.3 | 1.53 | 8.95 | 1.6 | 4.21 | 0.52 | 3.24 | 0.4 | 632.81 | 600.07 | 32.74 | 27.97 | 54.38 | 12.18 | 3.08 | 13.24 | 1.05 | 1.02 |
| Mean | 108.32 | 175.47 | 16.96 | 56.78 | 7.85 | 2.25 | 5.88 | 0.76 | 4.49 | 0.81 | 2.29 | 0.3 | 1.89 | 0.25 | 385.01 | 368.11 | 16.71 | 22.03 | 38.63 | 8.69 | 2.52 | 10.58 | 1.01 | 0.96 |
| Jolotca River | | | | | | | | | | | | | | | | | | | | | | | | |
| 1J | 52.4 | 86.3 | 9.54 | 35.3 | 5.96 | 1.1 | 4.59 | 0.63 | 3.96 | 0.78 | 2.38 | 0.34 | 2.45 | 0.34 | 206.07 | 190.6 | 15.47 | 12.32 | 14.45 | 5.53 | 1.52 | 5.03 | 0.64 | 0.9 |
| 4J | 55.6 | 95.9 | 10 | 34.2 | 6.02 | 1.1 | 4.81 | 0.72 | 4.39 | 0.84 | 2.69 | 0.39 | 2.97 | 0.4 | 220.03 | 202.82 | 17.21 | 11.79 | 12.65 | 5.81 | 1.31 | 4.02 | 0.62 | 0.95 |
| 8J | 72.3 | 113.5 | 11.65 | 40.5 | 6.07 | 1.3 | 4.65 | 0.64 | 4 | 0.77 | 2.33 | 0.34 | 2.28 | 0.33 | 260.64 | 245.32 | 15.33 | 16 | 21.43 | 7.5 | 1.65 | 6.20 | 0.74 | 0.92 |
| 11J | 94.7 | 155.5 | 16.9 | 60 | 9.11 | 2.17 | 7 | 0.92 | 5.75 | 1.06 | 3.04 | 0.41 | 2.67 | 0.37 | 359.61 | 338.38 | 21.23 | 15.94 | 23.97 | 6.54 | 2.12 | 7.84 | 0.83 | 0.91 |
| 16J | 66.8 | 110 | 12.5 | 45.2 | 7.2 | 1.35 | 5.78 | 0.78 | 4.84 | 0.9 | 2.66 | 0.37 | 2.54 | 0.35 | 261.27 | 243.05 | 18.22 | 13.34 | 17.77 | 5.84 | 1.84 | 6.21 | 0.64 | 0.89 |
| 18J | 83.6 | 141.5 | 15.65 | 56.7 | 8.61 | 1.87 | 6.8 | 0.91 | 5.66 | 1.05 | 3.05 | 0.42 | 2.81 | 0.39 | 329.01 | 307.93 | 21.08 | 14.6 | 20.1 | 6.11 | 1.96 | 7.04 | 0.74 | 0.92 |
| 19J | 88.9 | 156.5 | 17.25 | 61.9 | 9.44 | 2.07 | 7.3 | 0.97 | 5.78 | 1.07 | 3.07 | 0.42 | 2.77 | 0.38 | 357.81 | 336.06 | 21.75 | 15.45 | 21.69 | 5.93 | 2.14 | 7.79 | 0.76 | 0.94 |
| 20J | 240 | 432 | 55.3 | 203 | 29.3 | 7.96 | 20.8 | 2.42 | 13.6 | 2.3 | 6.02 | 0.73 | 4.44 | 0.56 | 1018.43 | 967.56 | 50.87 | 19.02 | 36.53 | 5.16 | 3.8 | 15.95 | 0.99 | 0.88 |
| 21J | 152 | 285 | 34.9 | 132 | 19.75 | 5.19 | 14.85 | 1.83 | 10.7 | 1.87 | 5 | 0.63 | 3.89 | 0.51 | 668.11 | 628.84 | 39.27 | 16.01 | 26.4 | 4.84 | 3.09 | 11.84 | 0.93 | 0.92 |
| 23J | 120 | 224 | 26.1 | 101.5 | 15.4 | 4.09 | 11.7 | 1.42 | 8.48 | 1.5 | 3.99 | 0.51 | 3.15 | 0.4 | 522.23 | 491.09 | 31.14 | 15.77 | 25.74 | 4.9 | 3.01 | 11.24 | 0.93 | 0.94 |
| Min | 52.4 | 86.3 | 9.54 | 34.2 | 5.96 | 1.1 | 4.59 | 0.63 | 3.96 | 0.77 | 2.33 | 0.34 | 2.28 | 0.33 | 206.07 | 190.6 | 15.33 | 11.79 | 12.65 | 4.84 | 1.31 | 4.02 | 0.62 | 0.88 |
| Max | 240 | 432 | 55.3 | 203 | 29.3 | 7.96 | 20.8 | 2.42 | 13.6 | 2.3 | 6.02 | 0.73 | 4.44 | 0.56 | 1018.43 | 967.56 | 50.87 | 19.02 | 36.53 | 7.5 | 3.8 | 15.95 | 0.99 | 0.95 |
| Mean | 91.98 | 158.19 | 17.8 | 64.6 | 10.03 | 2.23 | 7.71 | 1.01 | 6.16 | 1.13 | 3.26 | 0.44 | 2.94 | 0.4 | 368.72 | 345.33 | 23.19 | 14.89 | 21.17 | 5.77 | 2.13 | 8.31 | 0.77 | 0.92 |

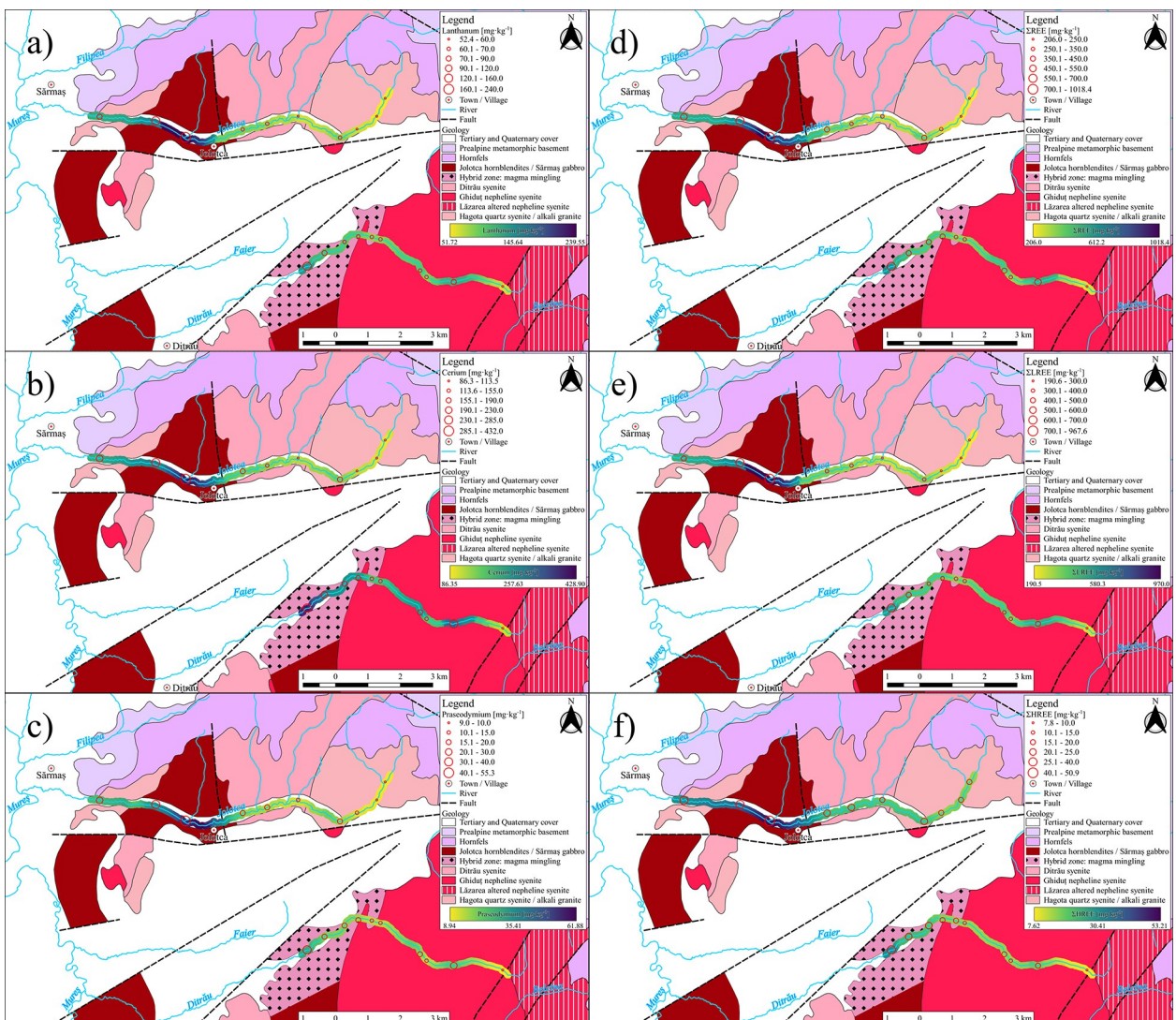

**Fig 3. Distribution maps of REE along Ditrău and Jolotca rivers.** a) Lanthanum; b) Cerium; c) Praseodymium; d) sum of REE; e) sum of Light REE; f) sum of Heavy REE.

In the petrography of DAM, Sr usually associates with Ca minerals, especially in syenites and hornblendites regions. The Sr/UCC concentration is 267 to 528 for syenites and 1179 for hornblendites [41]. Minerals with high content of Sr within the Massif are epidote, monazite, xenotime, hornblende, and apatite, with Sr concentrations that vary from 16195 for epidote and to 815 for hornblende [41]. We identified minerals such as hornblende and apatite in our sediment samples, confirming the presence of Sr, with concentrations higher than the UCC.

## 5.2 High-field strength element tracers

The normalized values of trace elements indicate higher enrichments of the high-field strength element tracers for both rivers than the UCC, especially for the Nb and Ta (Fig 4).

The Nb concentrations are much more enriched than the UCC, with a sample/UCC ratio higher from 13.8 to 26 for Ditrău samples (Fig 4A). In contrast, for the Jolotca River (Fig 4B), the ratio has an average of 15.8 and values that range from 10.3 to 24.3. Similarly, Ta has higher

**Table 5. Enrichment Factor values* for the REE contents in Ditrău and Jolotca rivers.**

| Sample ID | La | Ce | Pr | Nd | Sm | Eu | Gd | Tb | Dy | Ho | Er | Tm | Yb | Lu |
|---|---|---|---|---|---|---|---|---|---|---|---|---|---|---|
| Ditrău River | | | | | | | | | | | | | | |
| 1D | 16.47 | 12.62 | 11.44 | 8.66 | 5.85 | 7.50 | 6.61 | *4.81* | 5.36 | 5.04 | *4.35* | *4.37* | 6.84 | 5.34 |
| 6D | 33.55 | 24.66 | 25.16 | 19.74 | 13.53 | 18.31 | 16.08 | 11.08 | 11.39 | 9.73 | 7.17 | 6.38 | 8.98 | 6.49 |
| 9D | 15.38 | 12.23 | 12.41 | 10.37 | 7.82 | 11.41 | 10.22 | 7.30 | 7.55 | 6.33 | *4.83* | *4.29* | 5.96 | *4.44* |
| 10D | 17.28 | 12.85 | 12.68 | 10.30 | 7.36 | 10.32 | 8.66 | 6.17 | 6.35 | 5.34 | *4.05* | *3.71* | 5.08 | *3.87* |
| 15D | 20.71 | 16.03 | 16.05 | 13.18 | 10.12 | 14.36 | 13.12 | 9.24 | 9.58 | 8.01 | 5.99 | 5.51 | 7.60 | 5.96 |
| 16D | 19.24 | 15.43 | 15.75 | 12.94 | 10.09 | 14.17 | 13.10 | 9.40 | 9.96 | 8.48 | 6.34 | 5.74 | 7.90 | 5.97 |
| 17D | 24.36 | 18.58 | 18.57 | 15.16 | 10.98 | 16.07 | 14.17 | 10.04 | 10.32 | 8.75 | 6.55 | 5.76 | 7.91 | 5.78 |
| 18D | 19.60 | 17.27 | 17.16 | 13.84 | 12.00 | 17.73 | 15.53 | 11.95 | 11.39 | 9.34 | 7.06 | 6.15 | 9.12 | 5.97 |
| 20D | 19.31 | 16.92 | 17.21 | 13.95 | 12.35 | 18.47 | 16.17 | 12.31 | 11.76 | 9.67 | 7.20 | 6.60 | 9.08 | 6.05 |
| 22D | 22.76 | 19.21 | 21.76 | 19.19 | 15.16 | 22.31 | 19.94 | 13.89 | 14.01 | 11.71 | 8.31 | 7.15 | 9.80 | 6.72 |
| Jolotca River | | | | | | | | | | | | | | |
| 1J | 19.05 | 15.42 | 17.78 | 16.00 | 14.89 | 13.60 | 19.25 | 14.80 | 16.03 | 14.77 | 12.15 | 12.10 | 19.18 | 14.79 |
| 4J | 18.46 | 15.66 | 17.03 | 14.16 | 13.74 | 12.42 | 18.43 | 15.45 | 16.24 | 14.53 | 12.55 | 12.68 | 21.24 | 15.89 |
| 8J | 29.78 | 22.98 | 24.60 | 20.80 | 17.18 | 18.21 | 22.10 | 17.03 | 18.35 | 16.52 | 13.48 | 13.71 | 20.22 | 16.26 |
| 11J | 29.90 | 24.14 | 27.35 | 23.62 | 19.77 | 23.29 | 25.49 | 18.76 | 20.22 | 17.43 | 13.48 | 12.67 | 18.15 | 13.97 |
| 16J | 6.18 | 5.01 | 5.93 | 5.22 | *4.58* | *4.25* | 6.17 | 4.66 | 4.99 | *4.34* | *3.46* | *3.35* | 5.06 | *3.88* |
| 18J | 25.72 | 21.40 | 24.68 | 21.75 | 18.20 | 19.56 | 24.13 | 18.08 | 19.39 | 16.83 | 13.18 | 12.65 | 18.61 | 14.35 |
| 19J | 23.12 | 20.01 | 23.00 | 20.08 | 16.87 | 18.31 | 21.90 | 16.30 | 16.74 | 14.50 | 11.21 | 10.69 | 15.51 | 11.82 |
| 20J | 23.87 | 21.12 | 28.20 | 25.18 | 20.03 | 26.92 | 23.86 | 15.55 | 15.07 | 11.92 | 8.41 | 7.11 | 9.51 | 6.66 |
| 21J | 20.96 | 19.32 | 24.67 | 22.70 | 18.72 | 24.33 | 23.62 | 16.30 | 16.43 | 13.43 | 9.68 | 8.50 | 11.55 | 8.41 |
| 23J | 21.45 | 19.68 | 23.92 | 22.62 | 18.92 | 24.85 | 24.12 | 16.39 | 16.88 | 13.97 | 10.01 | 8.92 | 12.12 | 8.55 |

* Values are classified according to [78]: values in italic—moderate EF; values in bold—significant EF; values in bold and underlined—very high EF.

concentrations than UCC, with ratio values ranging from 5.6 to 15, with a mean of 8.9 for Ditrău (Fig 4A) and 10.9 to 21 for Jolotca (Fig 4B).

This enrichment of Nb and Ta in the sediments is related to the petrography of the area. In the case of DAM, these two elements do not form minerals of their own but associate with Ti and Zr minerals. The Pearson correlation (Fig 5) also points out that Nb has a strong correlation of 0.94 with Zr in the samples from the Jolotca River and a lower correlation of 0.64 in the Ditrău samples.

Ta has a correlation of 0.9 with Zr in the samples from Jolotca River and 0.36 in the Ditrău sample sediments. Compared to the UCC, the Zr/UCC ratio varies from 1.1 to 2 with an average of 1.4 in the sediment samples from Ditrău River, while for the Jolotca samples, the ratio has lower values from 0.8 to 2.2. The concentration of Zr in the samples from both rivers comes mainly from the presence of zirconium in the petrography of the DAM. Zr and Ti are both important indicators for heavy minerals. For both rivers, the sediment samples indicate the presence of heavy minerals such as titanite and zircon, also supported by the Zr and Ti values from the ICP-MS analysis plotted against the sum of REE, showing a heavy mineral enrichment trend (Fig 6).

## 5.3 Transitional metals

The analysis of this subgroup minerals, compared to the UCC, exhibit similar values, except for Hf, Ni, W, and Zn, which have mean values close to 1. For instance, the Hf/UCC ratio

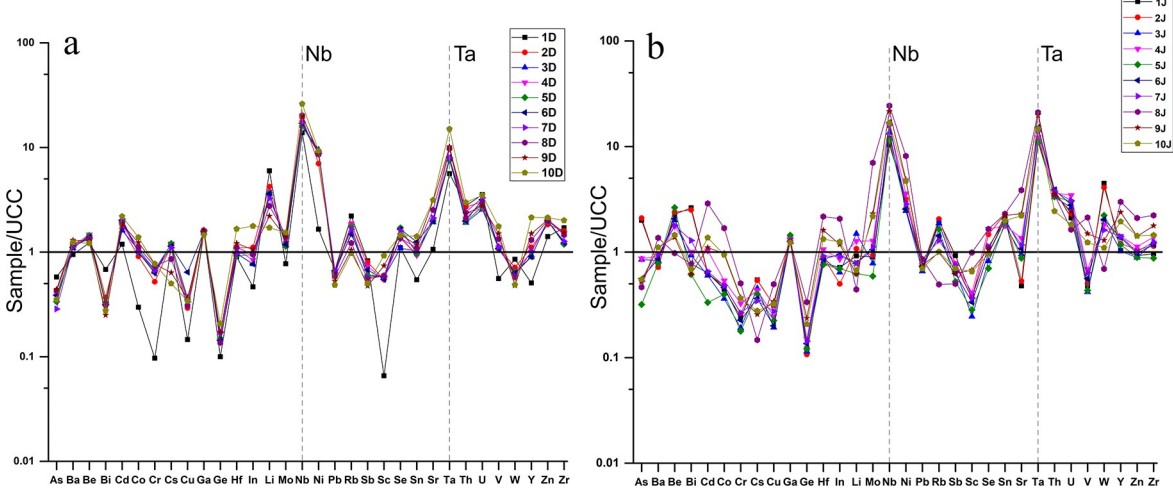

**Fig 4. Trace elements of river bed sediments normalized to the average values of the UCC.** a) Ditrău River, b) Jolotca River.

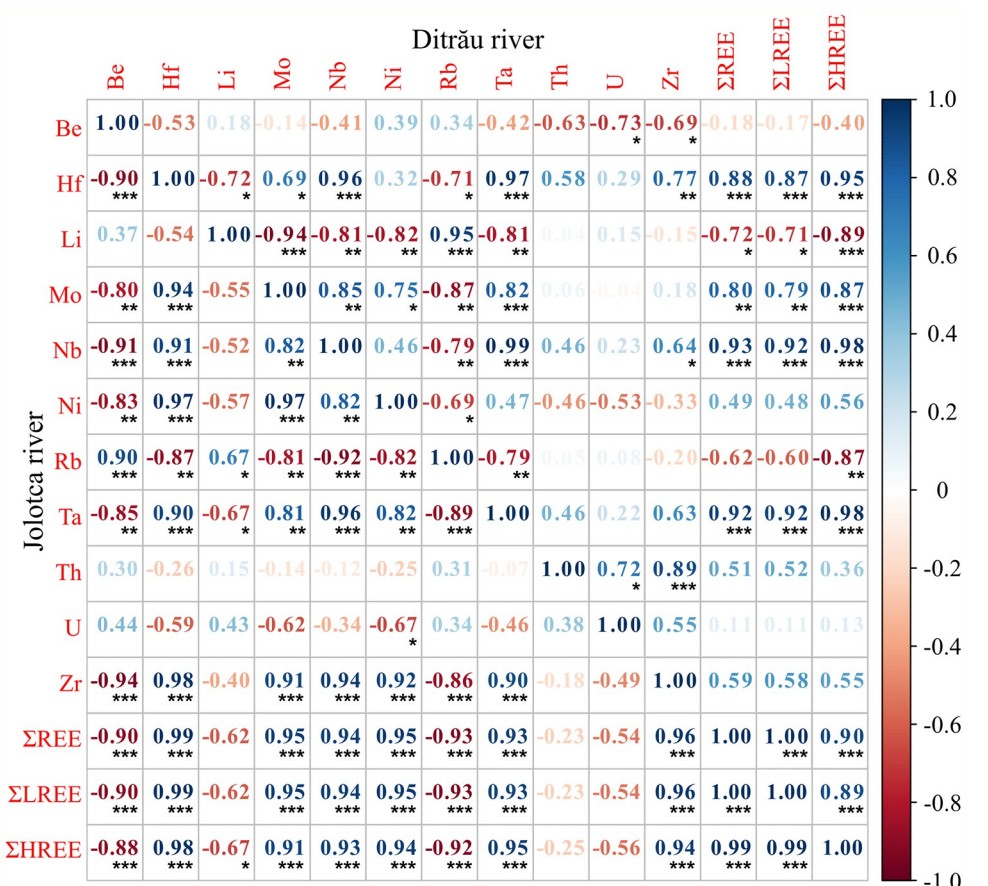

**Fig 5. Pearson correlation between trace elements and REE for Jolotca and Ditrău rivers.**

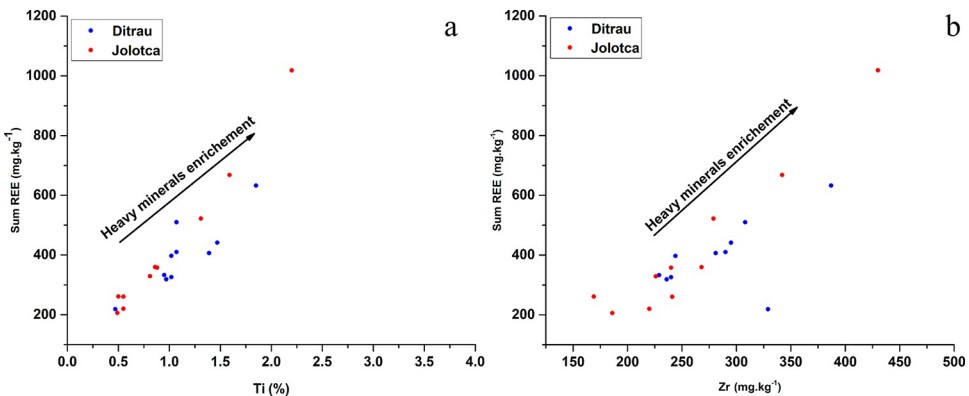

**Fig 6.** Heavy minerals enrichment of samples based on the variability in a) ΣREE vs. Ti pair and b) ΣREE vs. Zr pair.

varies from 0.9 to 1.6 for Ditrău samples, while for Jolotca sediment samples, the values range from 0.7 to 2.1. The high values of Zr concentrations in the DAM are associated with high Hf concentrations, as seen on different zircon minerals from red syenites with Hf concentrations ranging from 2640 mg·kg$^{-1}$ to 5907 mg·kg$^{-1}$ [41]. Hf is also found in other minerals, such as microcline, biotite, hornblende, apatite, and albite, with concentrations varying from 4 mg·kg$^{-1}$ to 22 mg·kg$^{-1}$ identified in the sediments we analyzed. As a possible indicator for the sediment source, Hf may be used to discriminate between different sources of the rocks (Fig 7).

Ni, compared to the UCC, varies from 1.6 to 9.4 with an average of 7.9 for the samples from the Ditrău River and from 2.4 to 8.1 with an average of 3.7 for the Jolotca River. This enrichment in Ni may be related to basic and ultrabasic rocks, which have high Ni concentrations in hornblendites and diorites.

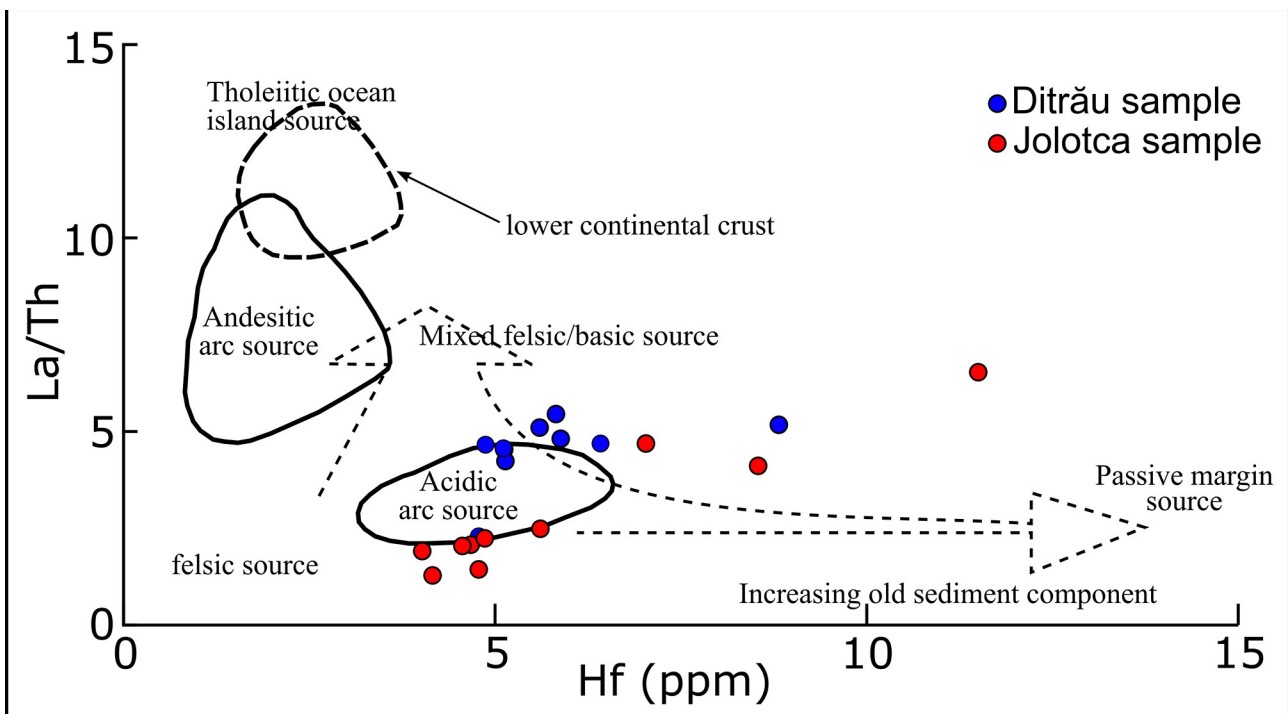

**Fig 7. Source and compositional discrimination diagram for our study area (modified after [83]).**

In the sediment samples from Ditrău River, W does not show any enrichment when compared to the UCC, but when looking at the samples from Jolotca River, the W/UCC ratio varies from 0.6 to 4.4 with an average of 2.1. This enrichment only occurs in the samples from the Jolotca River. It may be related to the association of tungsten with molybdenum mineralization [84], as there are known mineralizations of Mo in the Jolotca area [85].

When comparing Zn with the UCC, sediment samples from the Jolotca River show no enrichment trend with values close to the UCC. For the Ditrău River, the Zn/UCC ratio shows values that vary from 1.4 to 2.1, with an average of 1.9. This enrichment in Zn only for the sediment samples from the Ditrău River comes from the presence of the hornblendites and diorites [85].

## 5.4 Metalloids

The analyzed elements of this sub-group (As, Ge, and Sb) show typical values compared to the UCC. Ge shows a decreasing trend for the sediment samples from both rivers. The first two samples from Jolotca River present an anomaly for As with an As/UCC ratio of 2, possibly related to the presence of minerals such as pyrite.

## 5.5 Other metals/non-metals

For this sub-group, Ga, In, Se, and Pb show normal values compared to the UCC, except Bi and Sn. Bi shows depletion trends for the sediment samples from the Ditrău River and typical concentrations for the Jolotca River, but for the first two samples from this river, Bi shows an anomaly with a Bi/UCC ratio of 2.5. This anomaly can be accounted for by the presence in the Jolotca region of some veins with Bi minerals such as bismutine, and joseit [41].

Sn shows normal values for the sediment samples from Ditrău River compared to the UCC, but for the samples from Jolotca River, the Sn/UCC ratio varies from 1.7 to 2.3 with an average of 2. This enrichment in Sn in the samples from the Jolotca River is because the Sn appears in the DAM only in hornblendites and in some minerals such as zircon and pyrochlore, areas that are drained only by the Jolotca River.

## 5.6 U and Th

U shows high values for both rivers. In the case of Ditrău samples, the U/UCC ratio varies from 2.5 to 3.5, with an average of 3, and for the samples from Jolotca River, it varies from 1.6 to 3.4, with an average of 2.5. The Th/UCC ratio varies from 1.9 to 2.9, with an average of 2.3 in the samples from Ditrău River, and for Jolotca, it varies from 2.4 to 3.8, with an average of 3.4. There is a clear enrichment trend in both rivers for U and Th due to minerals such as xenotime and monazite, which contain traces of U and Th. Moreover, the area is characterized by syenites and nepheline syenites, previously described by [64].

## 5.7 Eu and Ce anomaly

Eu and Ce anomalies, defined as Eu/Eu* and Ce/Ce*, respectively, due to their distinct valence and radius, were calculated according to [86]. $Eu^{2+}$ more often behaves separately from the rest of REE ions and is reflected in the chondrite-normalized diagrams because Eu plots in a position notably different from that resulting from the interpolation between Sm and Gd. The Eu anomaly is numerically expressed as the ratio:

$$Eu_n/Eu^* \tag{Eq2}$$

where Eu* is:

$$\sqrt{Sm_n \cdot Gd_n} \qquad \qquad (Eq3)$$

If the ratio is >1, the anomaly is positive, while the ratio is <1, the anomaly is negative.

Ce anomaly occurs when Ce is partially oxidized to $Ce^{4+}$. It also can be positive or negative and is numerically expressed as the ratio:

$$Ce_n / Ce^* \qquad \qquad (Eq4)$$

where Ce* is:

$$\sqrt{La_n \cdot Pr_n} \qquad \qquad (Eq5)$$

While the intensity of the Ce anomaly and the slope $Nd_n/Yb_n$ are good indicators of the freshness of the lithogenic supplies, the direction and intensity of the Eu anomaly could help reveal the nature of the lithogenic supplies [87], considering the imprint of lithogenic supplies as significant if $Ce/Ce^* > 0.2$ and $Nd_n/Yb_n > 0.2$, as is the case for the analyzed samples in this study. Following the general statements of the dissolved oceanic REE behavior [88–90], the freshest lithogenic supplies are tagged by the weakest Ce anomaly and $Nd_n/Yb_n$ slope. According to [91], the La/Th and Hf distinction plots can distinguish between various source compositions and basic or elongated source composition indicators [92]. Derived felsic components have low or uniform La/Th values (<5) and Hf (3–7 ppm) concentrations [83]. Samples from the Ditrău area fall predominantly into the mixed felsic/basic source area, while the samples from the Jolotca area fall into the acidic source area (Fig 7).

In our case, the Eu anomaly shows an average of 1.01, with a minimum value of 0.96 and a maximum value of 1.04 for the Ditrău River. For the samples from Jolotca River, the Eu anomaly shows an average of 0.78, with a minimum value of 0.62 and a maximum of 0.98. Similarly, Ce anomaly averages 0.95 for Ditrău samples, with a minimum value of 0.91 and a maximum of 1.01. For Jolotca, the Ce anomaly has an average of 0.91, with a minimum value of 0.87 and a maximum of 0.95.

Hence, in the sediment samples from Ditrău, there is no Eu anomaly, and for some samples, the ratio is even below the threshold. The Eu anomaly is very weak for the samples from Jolotca River, with values lower than the threshold, with 0.62 for one sample. The same case occurs for the Ce anomaly, with both river samples having weak negative anomalies.

## 5.8 Rare earth elements

REE analysis have been a subject of interest for many authors regarding the DAM. In this case, the mean value of REE is much higher than the other averages from Romania, Europe, and other parts of the world (Table 6).

**5.8.1 REE normalization to chondrites and UCC.**   In our study, there is a clear enrichment in REE, especially in LREE (Fig 3). The samples from Ditrău River show an average ΣREE of 399.5 mg·kg$^{-1}$, and the samples from Jolotca show an average ΣREE of 420 mg·kg$^{-1}$, while the average ΣREE for the UCC is 148.1 mg·kg$^{-1}$ [80].

The average ΣLREE for the samples from Ditrău is 381.7 mg·kg$^{-1}$, and for the samples from Jolotca is 395.1 mg·kg$^{-1}$. Compared to the average ΣLREE of the UCC, 133.8 mg·kg$^{-1}$, we can see a clear enrichment trend in LREE for the samples from both rivers. When plotting the REE against the UCC (Fig 8), the enrichment trend for LREE and depletion for some samples in HREE is noticeable, but most of them still show similar values for HREE when compared to the UCC.

**Table 6. Different REE concentrations in river sediments from different areas and our study.**

| Element | La | Ce | Pr | Nd | Sm | Eu | Gd | Tb | Dy | Ho | Er | Tm | Yb | Lu |
|---|---|---|---|---|---|---|---|---|---|---|---|---|---|---|
| Our study | 99.8 | 166.6 | 17.4 | 60.56 | 8.87 | 2.24 | 6.74 | 0.88 | 5.26 | 0.96 | 2.73 | 0.36 | 2.36 | 0.31 |
| Mongolia [27] | 2.79 | 6.69 | 0.72 | 2.78 | 0.56 | 0.13 | 0.4 | 0.08 | 0.39 | 0.08 | 0.24 | 0.03 | 0.21 | 0.03 |
| Angola [93] | 43.2 | 85.3 | 9.6 | 34.6 | 5.85 | 1.21 | 4.21 | 0.63 | 3.59 | 0.72 | 2.18 | 0.35 | 2.5 | 0.4 |
| Romania [81] | 34.3 | 58.9 | 6.5 | 26.9 | 5.2 | 0.9 | 4.9 | 0.7 | 4.3 | 0.8 | 2.5 | 0.3 | 2.7 | 0.3 |
| Europe [82] | 41 | 83 | 9.2 | 36.6 | 6.9 | 1.1 | 6.3 | 0.9 | 5.4 | 1 | 3.1 | 0.4 | 3 | 0.4 |
| UCC [80] | 31 | 63 | 7.1 | 27 | 4.7 | 1 | 4 | 0.7 | 3.9 | 0.8 | 2.3 | 0.3 | 2 | 0.3 |

We observe that the samples taken from the Ditrău River have a predominance of "significant" EF with values between 5 and 20, and for the samples taken from the Jolotca River, the EF values fall mostly in the "very high" class, with values between 20 and 40 (Table 5). Overall, we can argue that the majority of all REE originate from anthropogenic processes, especially in the Jolotca area, where former mining activities took place.

Regarding the chondrite-normalized REE patterns [86], the sediment samples display a similar trend, an enrichment in LREE and slow depletion in HREE but not below the average for chondrites (Fig 8).

The fractioning chondrite normalized REE patterns for the Ditrău samples show an average $La_n/Yb_n$ of 38.5, and for the Jolotca samples, an average $La_n/Yb_n$ of 23.14. Comparable, the UCC/Chondrite shows an average $La_n/Yb_n$ of 10.4, pointing out the high enrichment of REE in our study area.

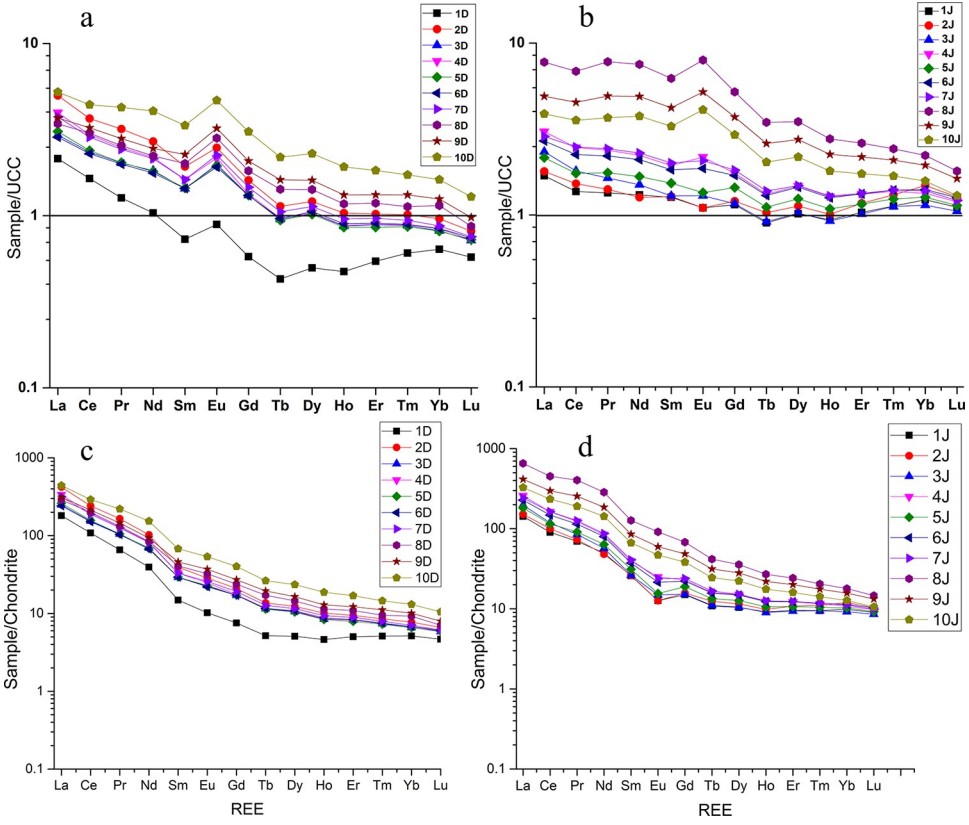

**Fig 8.** REE normalization to the UCC and to Chondrites for the samples from Ditrău (a, c) and Jolotca (b, d).

The fractioning chondrite normalized REE patterns for the Ditrău samples show an average $La_n/Yb_n$ of 38.5, and for the Jolotca samples, an average $La_n/Yb_n$ of 23.14. Comparable, the UCC/Chondrite shows an average $La_n/Yb_n$ of 10.4, pointing out the high enrichment of REE in our study area.

LREE fractionation ratio $La_n/Sm_n$ for the Ditrău samples shows an average of 8.3, and for Jolotca, an average of 5.5 when compared to the $La_n/Sm_n$ ratio of the UCC/Chondrite, which is 4.15. The ratio value of $La_n/Sm_n$ for the sediment samples from Jolotca River is closer to the UCC than the samples from Ditrău, which show a higher enrichment. The HREE fractionation ratio of $Gd_n/Yb_n$ for the Ditrău sediment samples has an average of 2.6, and for the Jolotca samples has an average of 2.3, slightly higher compared to the $Gd_n/Yb_n$ ratio of the UCC that is 1.6, indicating a light enrichment.

**5.8.2 REE sources.** DAM is known for high REE concentrations and therefore for REE mining interest, with mineralizations of REE identified in minerals such as xenotime, monazite, enriched REE veins in apatite [44], or accumulations of REE identified in rocks such as white syenites, diorites, hornblendites, red syenites and nepheline syenites [41].

Without intense weathering, river bed sediments can preserve REE patterns and be used to trace to the source rocks [94]. In our study, there is a strong correlation between the heavy minerals and the high concentrations of REE due to the presence of minerals such as zircon and titanite in the sediment samples from both rivers (Fig 5). While the intensity of the Ce anomaly and the slope $Nd_n/Yb_n$ are good indicators of the freshness of the lithogenic supplies, the direction and intensity of the Eu anomaly could help reveal the nature of the lithogenic supplies [87], considering the imprint of lithogenic supplies as significant if $Ce/Ce^* > 0.2$ and $Nd_n/Yb_n > 0.2$, as is the case for the analyzed samples in this study (Fig 9).

The variation of REE concentrations between the two rivers is also argued based on the Hf discrimination diagram (Fig 7), La/Yb vs ΣREE plot (Fig 9), and the distribution maps of the ΣLREE and ΣHREE (Fig 3). Jolotca River has higher REE concentrations due to the river's drainage basin, which has a more considerable petrographical variety than Ditrău.

Another strong correlation exists between Ca, P, and REE (Fig 10) considering that high REE concentrations exist in P and Ca minerals such as monazite, xenotime, titanite, and hornblende, previously mentioned also by [44].

# 6. Conclusions

In this work, we used the river bed sediments sampled from the Jolotca and Ditrău rivers to determine their geochemical composition, explicitly focusing on REE, and get more significant data about the geochemistry of the Massif. We identified and analyzed the presence of the REE and the content of heavy minerals and trace elements. In the riverbed sediments from DAM, the whole range of REE, from La to Lu, was identified in the bearing minerals of the area, such as Monazite and Epidote. The analysis revealed that LREE, particularly Cerium (Ce) and Lanthanum (La), are found in concentrations exceeding the UCC's. For instance, Cerium concentrations have values of 175.47 mg·kg$^{-1}$ in the Ditrău samples, while Lanthanum reached 108.32 mg·kg$^{-1}$, both of which are over twice the UCC reference values. Regarding mineralogical composition, Quartz, K Feldspar, and Albite are found across the river sediments, though Diopside appeared exclusively in Jolotca sediments, and Plagioclase was unique to Ditrău. The presence of trace elements like Zirconium, Niobium, and Tantalum also stood out, with the sediments showing high concentrations of Zirconium at 265.62 mg·kg$^{-1}$ and Niobium at 200.24 mg·kg$^{-1}$. These enrichments indicate a complex geochemical background influenced by both natural geological processes and anthropogenic activities.

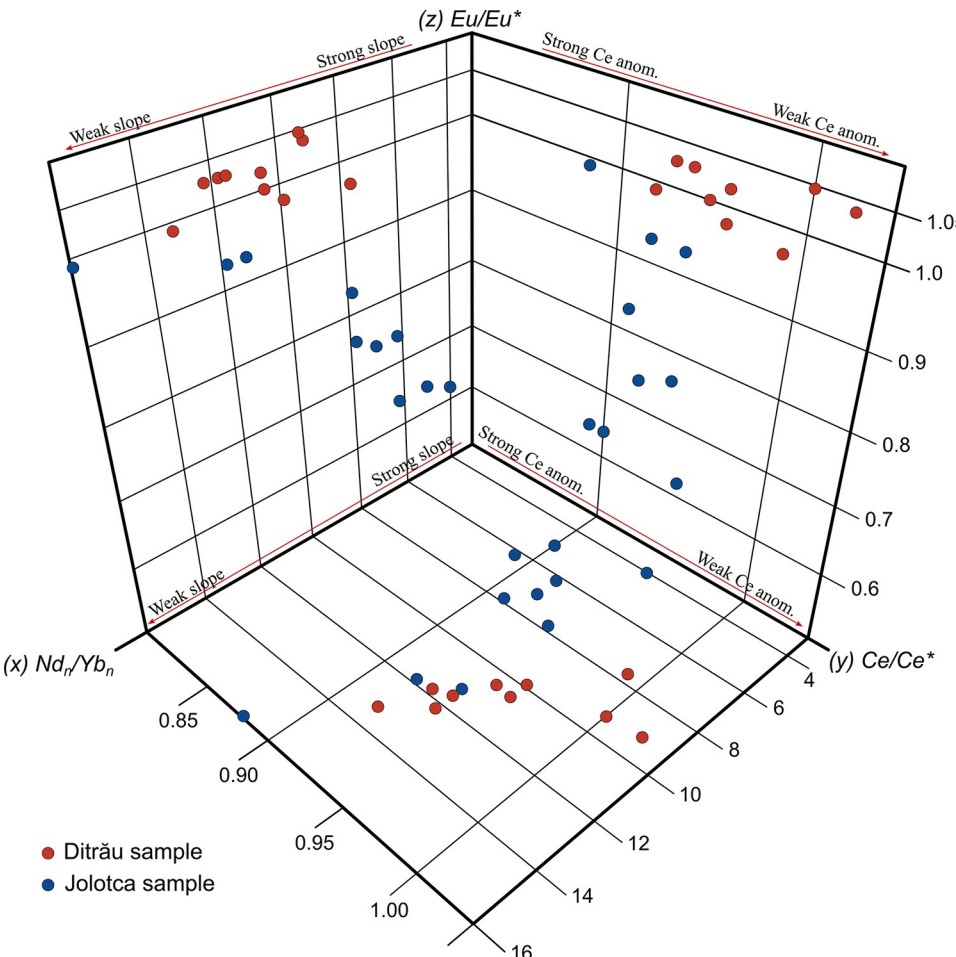

**Fig 9. Inter-correlation of the REE fractionations pointing out the freshness and the nature of the lithogenic supplies.** xy plan: $Nd_n/Yb_n = f(Ce/Ce^*)$; xz plan: $Nd_n/Yb_n = f(Eu/Eu^*)$; yz plan: $Ce/Ce^* = f(Eu/Eu^*)$.

The EF values in the Jolotca sediments are higher than those in Ditrău, particularly in the range of 20–40 of the "very high" class for Jolotca, compared to the "significant" class range of 5–20, which are in Ditrău samples. This disparity likely reflects the influence of historical mining activities near the Jolotca River, which might have contributed to the higher concentrations of REE. The Ce anomaly and the $Nd_n/Yb_n$ ratio that we used as indicators of lithogenic supplies indicate values well above the threshold for significant lithogenic input for both rivers, further underscoring the presence of fresh geological material in the sediment supply. Also, the La/Th and Hf distinction plots provide significant insights into the geological origin of the sediments. These plots indicate a mixed felsic/basic source for the Ditrău area, while the riverbed sediments from the Jolotca River point to an acidic source.

This study confirms the abundance of REE in the Ditrău and Jolotca rivers, which derive from both natural lithogenic sources, but also highlights the impact of the anthropogenic activities, particularly in areas like Jolotca, where past mining operations have left a significant imprint on the local environment. Also, the differentiation in sediment origin highly contributes to the understanding of the geological history of the massif and its geochemical composition. This geochemical analysis enriches our understanding of the Ditrău Alkaline Massif and

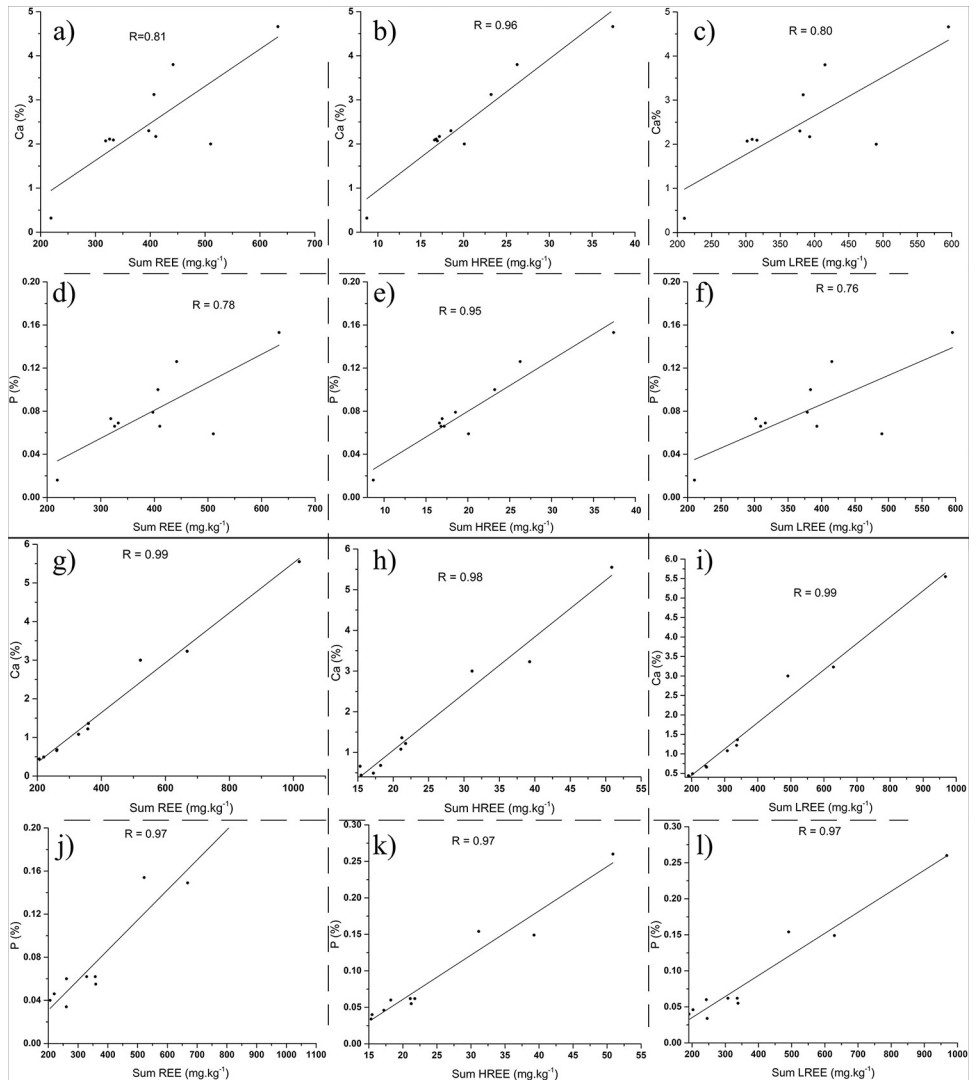

**Fig 10. Correlation plots of ΣREE, ΣHREE, and ΣLREE with Ca and P for the analyzed samples.** a, b, and c) correlation plots of ΣREE, ΣHREE, and ΣLREE with Ca for Ditrău samples; d, e, and f) correlation plots of ΣREE, ΣHREE, and ΣLREE with P for Ditrău samples; g, h, and i) correlation plots of ΣREE, ΣHREE, and ΣLREE with Ca for Jolotca samples; j, k, and l) correlation plots of ΣREE, ΣHREE, and ΣLREE with P for Jolotca samples.

provides crucial data on the environmental impacts of human activity in the region, which should further be evaluated and monitored.

## Acknowledgments

The authors thank two anonymous reviewers for their critical and insightful comments and observations, which helped to improve the article.

## Author Contributions

**Conceptualization:** Valentin Nicolae Coţac, Ovidiu Gabriel Iancu.

**Data curation:** Valentin Nicolae Coţac.

**Formal analysis:** Valentin Nicolae Coţac, Marius Cristian Sandu, Aurelia Andreea Loghin, Ovidiu Chişcan, George Stoian.

**Methodology:** Valentin Nicolae Coţac, Ovidiu Gabriel Iancu, Nicuşor Necula, Marius Cristian Sandu, Aurelia Andreea Loghin, Ovidiu Chişcan, George Stoian.

**Software:** Ovidiu Gabriel Iancu, Nicuşor Necula.

**Supervision:** Ovidiu Gabriel Iancu.

**Validation:** Ovidiu Gabriel Iancu, Marius Cristian Sandu.

**Visualization:** Nicuşor Necula.

**Writing – original draft:** Valentin Nicolae Coţac, Nicuşor Necula.

**Writing – review & editing:** Valentin Nicolae Coţac, Ovidiu Gabriel Iancu, Nicuşor Necula, Marius Cristian Sandu.

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
