## [Decision Letter · Decision Letter 0]

5 Aug 2024

PONE-D-24-23158Rare Earth Elements distribution in the river sediments of Ditrău Alkaline Massif, Eastern CarpathiansPLOS ONE

Dear Dr. Necula,

Thank you for submitting your manuscript to PLOS ONE. After careful consideration, we feel that it has merit but does not fully meet PLOS ONE’s publication criteria as it currently stands. Therefore, we invite you to submit a revised version of the manuscript that addresses the points raised during the review process.

We look forward to receiving your revised manuscript.

Kind regards,

Wajid Ali

Academic Editor

PLOS ONE

Journal Requirements:

3. Thank you for stating the following financial disclosure: "NN and AAL are grateful to Romanian Ministry of Research, Innovation and Digitization, within Program 1 – Development of the national RD system, Subprogram 1.2 – Institutional Performance – RDI excellence funding projects, Contract no.11PFE/30.12.2021, for financial support.

NN has used the computational facilities given by the infrastructure support from the Operational Program Competitiveness 2014–2020, Axis 1, under POC/448/1/1 Research infrastructure projects for public R&D institutions/Sections F 2018, through the Research Center with Integrated Techniques for Atmospheric Aerosol Investigation in Romania (RECENT AIR) project, under grant agreement MySMIS no. 12732."

Please state what role the funders took in the study.  If the funders had no role, please state: ""The funders had no role in study design, data collection and analysis, decision to publish, or preparation of the manuscript."

4. We note that your Data Availability Statement is currently as follows: "All relevant data are within the manuscript and its Supporting Information files."

5. We note that Figures 1,2 and 3 in your submission contain [map/satellite] images which may be copyrighted. All PLOS content is published under the Creative Commons Attribution License (CC BY 4.0), which means that the manuscript, images, and Supporting Information files will be freely available online, and any third party is permitted to access, download, copy, distribute, and use these materials in any way, even commercially, with proper attribution. For these reasons, we cannot publish previously copyrighted maps or satellite images created using proprietary data, such as Google software (Google Maps, Street View, and Earth). For more information, see our copyright guidelines: http://journals.plos.org/plosone/s/licenses-and-copyright.

1. You may seek permission from the original copyright holder of Figures 1,2 and 3 to publish the content specifically under the CC BY 4.0 license.  

Reviewers' comments:

Reviewer's Responses to Questions

**Comments to the Author**

1. Is the manuscript technically sound, and do the data support the conclusions?

Reviewer #1: Yes

Reviewer #2: Yes

2. Has the statistical analysis been performed appropriately and rigorously? 

Reviewer #1: Yes

Reviewer #2: Yes

3. Have the authors made all data underlying the findings in their manuscript fully available?

Reviewer #1: Yes

Reviewer #2: Yes

4. Is the manuscript presented in an intelligible fashion and written in standard English?

Reviewer #1: Yes

Reviewer #2: Yes

5. Review Comments to the Author

Reviewer #1: The manuscript titled “Rare Earth Elements distribution in the river sediments of Ditrău Alkaline Massif, Eastern Carpathians” by e Coțac et al. investigated the distribution characteristics of REEs in the river sediments based on the ICPMS, powder X-ray diffractometry, and electronic microscopy. Some interesting findings have been concluded. However, authors did not mention the quality control in the methods, making the results doubtful. The specific comments were as follows:

1. Line 41. I think “trace elements” can be deleted.

2. Line 68-69. I think the specific influences of environmental processes on the chemical composition of sediments can be introduced.

3. Line 70-72. These descriptions seem repetitive with previous context.

4. Authors need to add more introduction about REE in tracing lithogenic processes in sediments.

5. In the manuscript, “rare earth elements” should be written as “REE” because they have been defined in the manuscript.

6. Line 146 and 155, repeated descriptions.

7. Line 3.2. For the chemical analysis, the quality assurance is missing.

8. Line 248. “ICP-MS”, is that true?

9. Line 351. Authors should indicate which baseline they used to calculate the anomalies.

10. Line 361. As authors discussed the element enrichment in this section, the enrichment factors for these elements can be calculated, which can provide more intuitive comparisons for them.

11. Although authors made some comparisons between the measured elemental concentrations and UCC, the influences of various lithogenic processes on the element enrichment were not well discussed.

12. Figure 1 should be modified. We cannot identify where the sampling sites were located.

Reviewer #2: Manuscript Number: PONE-D-24-23158_comments

Comments

The manuscript having ID PONE-D-24-23158 and entitled “Rare Earth Elements distribution in the river sediments of Ditrău Alkaline Massif, Eastern Carpathians” has focused on the most important environmental geosciences issue. However, the manuscript needs minor revision before recommending for publication in the Journal of PLOS One.

Detail comments are attached.

6. PLOS authors have the option to publish the peer review history of their article (what does this mean?). If published, this will include your full peer review and any attached files.

Reviewer #1: No

Reviewer #2: No

---

## [Author Response · Author response to Decision Letter 0]

1 Nov 2024

Response to reviewer comments

Reviewer #1:

The manuscript titled “Rare Earth Elements distribution in the river sediments of Ditrău Alkaline Massif, Eastern Carpathians” by Coțac et al. investigated the distribution characteristics of REEs in the river sediments based on the ICPMS, powder X-ray diffractometry, and electronic microscopy. Some interesting findings have been concluded. However, authors did not mention the quality control in the methods, making the results doubtful.

Thank you for your comments and suggestions. Regarding the quality control of the analytical methods, we extended the text in section 3.2 and detailed process for the analyses we carried on. Below, we answer the specific comments, point by point.

The specific comments were as follows:

1. Line 41. I think “trace elements” can be deleted.

Thank you for your suggestion. However, considering that we also have a section about trace elements in the paper, we think keeping this keyword will make the article much more visible to researchers who work on that specific topic.

2. Line 68-69. I think the specific influences of environmental processes on the chemical composition of sediments can be introduced.

Thank you for your suggestion. We significantly improved the introduction section by adding more background information about the lithogenic processes in sediments, the influences of environmental processes on the chemical composition of sediments, and the study's objectives. The text about the environmental processes on the chemical composition of sediments corresponds to lines 92-102.

3. Line 70-72. These descriptions seem repetitive with previous context.

Thank you for pointing this out. Indeed, the entire paragraph repeats the previous text. We removed this part from the manuscript.

4. Authors need to add more introduction about REE in tracing lithogenic processes in sediments.

Thank you for your suggestion. We significantly improved the introduction section by adding more background information about the lithogenic processes in sediments, the influences of environmental processes on the chemical composition of sediments, and the study's objectives. The importance of the REE in tracing lithogenic processes in sediments is well presented now, with appropriate references, and can be found in the introduction section between lines 59-80.

5. In the manuscript, “rare earth elements” should be written as “REE” because they have been defined in the manuscript.

Thank you for pointing this out. We checked the entire manuscript for REE, LREE, HREE acronyms and changed it accordingly. Also for UCC and DAM acronyms.

6. Line 146 and 155, repeated descriptions.

Thank you for pointing this out. We adapted the text and removed the repetitiveness of the ideas.

7. Line 3.2. For the chemical analysis, the quality assurance is missing.

Thank you for your comment. We enhanced the 3.2 section (Analytical methods), in which we included the quality control of the methods and the recovery ratios.

8. Line 248. “ICP-MS”, is that true?

Thank you for pointing this out. The answer is No. Initially, we converted the elements from the ICP-MS into oxides. However, to reduce the readers' confusion, we used the raw values of results from the ICP-MS analysis in the table for the revised version of the manuscript. Accordingly, we changed the text before the table, replacing the oxides values with the elements values. Also, we changed Figure 6 and used the values for Ti instead of TiO2.

9. Line 351. Authors should indicate which baseline they used to calculate the anomalies.

Thank you for your comment. Regarding Cerium and Europium anomalies, they were calculated according to Taylor and McLennan, 1985. This is clearly stated in the text now in line 519.

10. Line 361. As authors discussed the element enrichment in this section, the enrichment factors for these elements can be calculated, which can provide more intuitive comparisons for them.

Thank you for your suggestion. We calculated the enrichment factor (EF) of the elements and inserted specific text for this part and a new table in section 4.3.

11. Although authors made some comparisons between the measured elemental concentrations and UCC, the influences of various lithogenic processes on the element enrichment were not well discussed.

Thank you for your suggestion. We included in the manuscript a new paragraph which refers to the enrichment factor values that we derived for the REE and relate it to the anthropic activity in the area.

12. Figure 1 should be modified. We cannot identify where the sampling sites were located.

Thank you for pointing this out. We modified the first figure to improve the visibility of the sampling sites, indicated by larger red triangles.

Reviewer #2:

Detailed comments

The manuscript having ID PONE-D-24-23158 and entitled “Rare Earth Elements distribution in the river sediments of Ditrău Alkaline Massif, Eastern Carpathians” has focused on the most important environmental geosciences issue. However, the manuscript needs minor revision before recommending for publication in the Journal of PLOS One.

Dear reviewer, thank you for your time and consideration regarding our article. Below you will find the point-by-point answers to your detailed comments.

1. Abstract: this section need to be strengthen with prominent results

Thank you for your suggestion. We significantly changed the abstract and improved it by adding relevant and the most significant results in this section. To comply with the journal’s length requirements, we removed a chunck of information about the study area and replaced it with relevant results information.

2. Acronyms need to be explained when first time used. Then use the abbreviation consistently

Thank you for pointing this out. We checked the entire manuscript for acronyms and changed their abbreviations and explanations accordingly. However, we kept the abbreviations for the elements as we consider them universally known by readers.

3. Introduction, Lines 63-65: This part lacks recent background information and needs to be cited with the following most relevant and recent citations:

https://link.springer.com/article/10.1007/s11356-022-24160-9

https://www.sciencedirect.com/science/article/abs/pii/S2352186420316333

https://doi.org/10.1080/15320383.2024.2306485

https://doi.org/10.1007/s11368-023-03484-0

Thank you for your suggestion. We significantly improved the introduction section with background information about both the lithogenic processes in sediments and the influences of environmental processes on the chemical composition of sediments.

4. Justification of this study needs to be strengthened and provide clear objectives

Thank you for your suggestion. We improved the manuscript and clearly stated the objectives of the study in the introduction section. See lines 109-119 in the reviewed manuscript.

5. Material and methods: sediments sampling need to be explained in details. If the sediments samples were collected in July 2018, then why you wait till 2024 to publish this work?

Thank you for pointing out this aspect. However, in the text in section 3.1, we explained thoroughly how the samples were acquired in terms of location, choice of the sampling location, and sample size taken from the field. We also added one more sentence about the tools we used. Regarding the period when the samples were collected, we removed that part to reduce the confusion for the readers. To answer your question, the first author started to work in industry, having a demanding job on which he focused more, slowing down the process of preparing the samples in the laboratory. The pandemic also happened during this time, which made the process even more difficult.

6. Information about precision and accuracy is missing

This comment/suggestion is unclear. Are you referring to the quality assurance of the chemical analysis? In this case, we enhanced the 3.2 section (Analytical methods), in which we included the quality control of the methods and the recovery ratios. Regarding the distribution maps' accuracy, we already mentioned in section 3.3 (Geostatistical analysis) that we performed 1000 conditional simulations of the point data and used their mean to reduce the uncertainties of a few point samples.

7. Results and discussion: This section need to be strengthened with disucssion

Thank you for your suggestion. We significantly improved these sections, including the elements' enrichment factor (EF) values and discussions about the Ce anomaly significance. We also pointed out the sediment origin based on the La/Th and Hf distinction plots.

The authors are indebted to the reviewers for their comments and observations, which led to a substantial improvement of the manuscript submitted for publication

---

## [Decision Letter · Decision Letter 1]

19 Nov 2024

Rare Earth Elements distribution in the river sediments of Ditrău Alkaline Massif, Eastern Carpathians

PONE-D-24-23158R1

Dear Dr. Necula,

We’re pleased to inform you that your manuscript has been judged scientifically suitable for publication and will be formally accepted for publication once it meets all outstanding technical requirements.

Kind regards,

Wajid Ali

Academic Editor

PLOS ONE

Additional Editor Comments (optional):

Reviewers' comments:

Reviewer's Responses to Questions

**Comments to the Author**

1. If the authors have adequately addressed your comments raised in a previous round of review and you feel that this manuscript is now acceptable for publication, you may indicate that here to bypass the “Comments to the Author” section, enter your conflict of interest statement in the “Confidential to Editor” section, and submit your "Accept" recommendation.

Reviewer #1: All comments have been addressed

Reviewer #2: All comments have been addressed

2. Is the manuscript technically sound, and do the data support the conclusions?

Reviewer #1: Yes

Reviewer #2: Yes

3. Has the statistical analysis been performed appropriately and rigorously? 

Reviewer #1: Yes

Reviewer #2: Yes

4. Have the authors made all data underlying the findings in their manuscript fully available?

Reviewer #1: Yes

Reviewer #2: Yes

5. Is the manuscript presented in an intelligible fashion and written in standard English?

Reviewer #1: Yes

Reviewer #2: Yes

6. Review Comments to the Author

Reviewer #1: The authors have made modifications according to the comments. The revised manuscript was improved and all the questions from reviewers were answered. I recommend that it can be accepted.

Reviewer #2: This manuscript has been revised and accordingly and recomended for publication in its current form.

7. PLOS authors have the option to publish the peer review history of their article (what does this mean?). If published, this will include your full peer review and any attached files.

Reviewer #1: No

Reviewer #2: No

---

## [Editor Report · Acceptance letter]

25 Nov 2024

PONE-D-24-23158R1 

PLOS ONE

Dear Dr. Necula, 

I'm pleased to inform you that your manuscript has been deemed suitable for publication in PLOS ONE. Congratulations! Your manuscript is now being handed over to our production team.

Kind regards, 

on behalf of

Dr. Wajid Ali 

Academic Editor

PLOS ONE